# CHANGANYA NDIMI: CODE-SWITCHED SPEECH GENERATION WITH A DIFFUSION PRIOR AND LINGUISTIC CONSTRAINTS

## ABSTRACT

We address code-switched speech generation by *editing* monolingual utterances with a pretrained diffusion-based speech model guided by linguistic constraints. Our method requires no parallel code-switched data. Instead, generation is conditioned on two differentiable modules—a multilingual language classifier and a contrastively trained segment encoder—that jointly guide where to insert semantically coherent, sociolinguistically appropriate foreign segments. During reverse diffusion, the system iteratively refines noisy speech representations, performing targeted segment substitutions while preserving fluency, prosody, and speaker identity.

On a semantically aligned corpus spanning five African languages from three language families, our approach achieves strong performance: segment-level COMET 0.815, LaBSE similarity 0.880, and 6.7% Equal Error Rate (EER) for speaker identity preservation. The model also reproduces natural code-switching patterns—frequency, temporal distribution, and alternation rates—without explicit supervision for such behaviors. To our knowledge, this is the first system to enable *controlled multilingual infusion within a single utterance*, highlighting guided diffusion as a flexible, plug-and-play framework for low-resource multilingual speech generation. Audio samples are available at: `https://github.com/codeSwitchLugha/CodeSwitch`.

## 1 INTRODUCTION

Code-switching—the fluid alternation between languages within an utterance—is widespread among African speakers Biswas et al. (2022); Sitaram et al. (2019). Yet most speech technologies (ASR, S2ST, SLU, speech LLMs) still rely mainly on monolingual data, as high-quality code-switched corpora are scarce. Code-switched speech synthesis remains relatively underexplored, and collecting spontaneous code-switched audio is costly and often yields unnatural speech Tarunesh et al. (2021); Hsu et al. (2023). We address this gap with a method that transforms monolingual corpora into realistic code-switched utterances via minimal, semantically coherent edits. Our approach samples from a constrained denoising diffusion model (DDPM): starting from noise, the model iteratively refines an utterance while two differentiable controllers guide generation:

(a) $c_1(x, y)$: a language-ID (LID) controller that decides *where* and *how much* to switch;

(b) $c_2(x, y)$: a multilingual segment encoder that determines *what* foreign-language content to insert by swapping in semantically matched segments.

We follow the *plug-and-play diffusion* paradigm: instead of retraining the generative model, we keep the diffusion prior $p(x)$ frozen and attach external constraint modules that steer sampling at test time. Formally,

$$p(x \mid y) \propto p(x)\, C(x, y), \qquad C(x, y) = c_1(x, y)\, c_2(x, y), \tag{1}$$

where $x$ denotes an utterance-level waveform and $y$ encodes the infusion specification (host/foreign language set, switch prior, optional source semantics, retrieval index). The prior $p(x)$ models natural speech (speaker identity, prosody), while $C(x, y)$ modulates *when/how much* to leave the host language ($c_1$) and *what* foreign content to insert ($c_2$). Guidance is implemented as a short, time-ramped penalty during sampling; the full derivation is given in App. B.

Experiments on four African languages plus English show that the synthesized speech is fluent, semantically aligned, and speaker-consistent according to automatic metrics (SacreBLEU, BERTScore, COMET, LaBSE, ECAPA-TDNN EER) and human judgments. The method can supply code-switched data for low-resource speech recognition, speech-to-speech translation, and multilingual LLM training. Our contributions are:

1. **Retrieval-augmented, plug-and-play diffusion** for code-switched speech that requires no parallel CS data and offers controllable *where/what* switching via external controllers on a frozen diffusion prior.

2. **Two complementary controllers**: LID-based switching ($c_1$) and retrieval-based semantic infusion ($c_2$) with late-commit and blend-and-write-back schedules.

3. **Evaluation suite** covering semantic fidelity, speaker consistency, prosody continuity at switch boundaries, code-switch structure, and human ratings.

4. **Empirical results on five languages** (Swahili, Luo, Kikuyu, Nandi, English) showing fluent, semantically aligned, speaker-consistent code-switching and downstream utility.

## 2 METHOD

### 2.1 PROBLEM FORMULATION

Our goal is to sample code-switched utterances from the constrained posterior in Eq. 1. Here, $x$ denotes a waveform and $y$ is an *infusion specification* that encodes how code-switching should occur. We write $y = (\mathcal{S}_{\text{inf}}, \pi_{\text{switch}}, \phi_{\text{src}}, \mathcal{R})$, where $\mathcal{S}_{\text{inf}}$ is the allowed foreign-language set, $\pi_{\text{switch}}$ is a prior over *where/how much* to switch, $\phi_{\text{src}}$ is a semantic representation of the host utterance, and $\mathcal{R}$ is a retrieval index over foreign-language segments. The constraint term $C(x, y)$ (Eq. 1) factorizes as $c_1(x, y)c_2(x, y)$, where $c_1$ encourages plausible switch locations given $\pi_{\text{switch}}$ and the LID model, and $c_2$ enforces semantic consistency between the edited utterance and the host. The prior $p(x)$ is implemented by a pretrained DDPM vocoder (Sec. 2.2), which ensures natural speaker identity, prosody, and acoustic realism.

Direct inference in $p(x \mid y)$ is intractable because the diffusion prior introduces a sequence of latent noise variables. We therefore augment $x$ with diffusion latents $h = \{x_1, \ldots, x_T\}$ and work with the joint model $p(x, h) = p(h)\, p(x \mid h)$, leading to the variational free-energy objective

$$F(q) = \text{KL}\big(q(x, h) \,\|\, p(x, h)\big) - \mathbb{E}_{q(x)}[\log C(x, y)], \qquad (2)$$

where $q(x, h)$ is a variational distribution over utterances and diffusion trajectories. The KL term favors utterances that are likely under the diffusion prior $p(x, h)$, while $-\mathbb{E}_{q(x)}[\log C(x, y)]$ steers samples toward satisfying the code-switching constraints. A step-by-step derivation of Eq. 2 from Eq. 1 and the DDPM joint model $p(x, h)$ is given in Appendix A.

### 2.2 FREE-ENERGY OBJECTIVE IN THE DDPM FRAMEWORK

We use a standard denoising diffusion probabilistic model (DDPM) Ho et al. (2020) as a frozen speech prior. The forward process gradually corrupts clean speech $x_0$ to noise, $q(x_t \mid x_0) = \mathcal{N}\big(x_t; \sqrt{\bar{\alpha}_t}\, x_0, (1 - \bar{\alpha}_t)I\big)$, $t = 1, \ldots, T$, and the reverse process is parameterized as $p_\theta(x_{t-1} \mid x_t) = \mathcal{N}\big(x_{t-1}; \mu_\theta(x_t, t), \Sigma_\theta(x_t, t)\big)$, $x_T \sim \mathcal{N}(0, I)$. As in Ho et al. (2020), maximizing $\log p_\theta(x_0)$ is equivalent (up to constants) to minimizing the noise-prediction loss

$$\mathcal{L}_{\text{DDPM}} = \mathbb{E}_{t, \epsilon_0}\Big[\|\hat{\epsilon}_\theta(x_t, t, M) - \epsilon_0\|_2^2\Big], \quad x_t = \sqrt{\bar{\alpha}_t}\, x_0 + \sqrt{1 - \bar{\alpha}_t}\, \epsilon_0, \qquad (3)$$

where $\epsilon_0 \sim \mathcal{N}(0, I)$ and $M$ denotes conditioning inputs (speaker, text, etc.).

To incorporate linguistic constraints without retraining the prior, we target the constrained posterior in Eq. 1, where $p(x)$ is the DDPM prior, $y$ encodes the infusion specification, and $c_1, c_2$ are our controllers (Sec. 1). Direct inference in Eq. 1 is intractable because $p(x)$ is defined via the latent diffusion trajectory $h = \{x_1, \ldots, x_T\}$. Following Chung et al. (2023), we adopt a mode-seeking variational family $q(x) = \delta(x - \eta)$, so that the free-energy objective in Eq. 2 reduces to a function of a single clean sample $\eta$.

Reparameterizing $x_t = \sqrt{\bar{\alpha}_t}\,\eta + \sqrt{1 - \bar{\alpha}_t}\,\epsilon_0$ and sampling $t \sim U(1, T)$, we obtain the practical plug-and-play objective used in our guided sampler:

$$F(\eta) = \mathbb{E}_{t,\epsilon_0}\left[\|\hat{\epsilon}_\theta(x_t, t, M) - \epsilon_0\|_2^2\right] - \log C(\eta, y), \qquad (4)$$

where the first term pushes $\eta$ toward high-likelihood speech under the prior and $-\log C(\eta, y)$ adds soft guidance from the switch and semantic controllers. In practice, this corresponds to standard DDPM reverse updates augmented with constraint gradients, realizing plug-and-play code-switching: the diffusion prior $p(x)$ remains frozen and the linguistic controllers steer generation at test time. Appendix B provides the complete derivation from the joint model $p(x, h)$ to Eq. 4.

# 3 DIFFUSION-BASED CODE-SWITCHING MODEL (DCSM)

## 3.1 CONSTRAINT $c_1(x, y)$: LANGUAGE IDENTIFICATION AND INFUSION DECISION

The first constraint $c_1(x, y)$ decides whether a *monolingual* host utterance $x_j$ should undergo foreign-language infusion, and if so, how strongly. Utterances that already contain substantial foreign material are left as they are, while mostly clean monolingual speech is treated as a candidate for code-switching.

**Segment-level foreignness.** We segment $x_j$ into $n$ short spans $\{s_{x_j}^{(i)}\}_{i=1}^n$ (e.g., log-Mel windows). Each span is fed to a frozen multilingual LID classifier $f_{\mathrm{cl}}$, which outputs a posterior over the host language $\ell_{\mathrm{mono}}$ and the infusion-eligible languages $\mathcal{S}_{\mathrm{inf}}(y) = \{\ell_1, \ldots, \ell_m\}$: $p(\ell \mid s_{x_j}^{(i)}) = f_{\mathrm{cl}}(s_{x_j}^{(i)})_\ell, \qquad \ell \in \mathcal{S}_{\mathrm{all}} = \{\ell_{\mathrm{mono}}\} \cup \mathcal{S}_{\mathrm{inf}}(y)$. For each segment we define a *foreignness score* $u^{(i)} = \sum_{\ell \in \mathcal{S}_{\mathrm{inf}}(y)} p(\ell \mid s_{x_j}^{(i)}) \in [0, 1]$, and aggregate these into an utterance-level measure

$$P_{\mathrm{foreign}}(x_j) = \frac{1}{n} \sum_{i=1}^n u^{(i)} \in [0, 1]. \qquad (5)$$

Here, $P_{\mathrm{foreign}}(x_j) \approx 0$ indicates that the classifier sees $x_j$ as strongly monolingual in $\ell_{\mathrm{mono}}$, while larger values signal that many segments already exhibit foreign-language characteristics (existing code-switches, borrowings, or noisy labels).

**Global controller and gating.** We use $P_{\mathrm{foreign}}(x_j)$ to control whether and how much the utterance should be edited. Given a target switch rate $\pi_{\mathrm{switch}} \in [0, 1]$, we define $c_1(x_j, y) = \sigma\big(\alpha_1\,[\,\pi_{\mathrm{switch}} - P_{\mathrm{foreign}}(x_j)\,]\big)$, with sharpness $\alpha_1 > 0$ and sigmoid $\sigma(\cdot)$. When the utterance is *less* foreign than the target ($P_{\mathrm{foreign}}(x_j) < \pi_{\mathrm{switch}}$), $c_1(x_j, y) \approx 1$ and the constraint encourages infusion; when it is already *more* foreign than desired, $c_1(x_j, y)$ shrinks toward 0 and suppresses further switching. For stricter control, we optionally use a hard gate $z_{\mathrm{infuse}} = \mathbb{1}\big(P_{\mathrm{foreign}}(x_j) \leq \tau\big)$, with tolerance $\tau \in [0, 1]$. Only utterances whose foreignness is below $\tau$ are considered for code-switching; those with $P_{\mathrm{foreign}}(x_j) > \tau$ are left untouched.

**Local soft masks for segment selection.** The same foreignness scores also provide soft, per-segment priorities for where infusion should happen. We define local gates $g^{(i)} = \sigma\big(\alpha_1\,[\,\pi_{\mathrm{switch}} - u^{(i)}\,]\big)$, so segments that look *more* monolingual (low $u^{(i)}$) receive higher weights $g^{(i)}$ and are preferred as candidates for replacement inside the infusion constraint $c_2$. At inference time, one may derive hard gates $z^{(i)} \in \{0, 1\}$ from $g^{(i)}$, while keeping gradients through the sigmoid during backprop.

**Contribution to the guided objective.** In the guided sampling objective, $c_1$ contributes via the scalar penalty

$$\mathcal{L}_{c_1}(x_j, y) = -\log c_1(x_j, y) = -\log \sigma\big(\alpha_1\,[\,\pi_{\mathrm{switch}} - P_{\mathrm{foreign}}(x_j)\,]\big), \qquad (6)$$

which is added to the guided objective in Eq. 4 with weight $\lambda_1$. This directly ties the LID-based controller to the diffusion trajectory: utterances that are too monolingual relative to $\pi_{\mathrm{switch}}$ are

nudged toward more foreign infusion, while utterances that are already highly foreign are protected from further editing. The classifier $f_{\mathrm{cl}}$ is pretrained and frozen during diffusion. It is trained to detect segment-level foreign-language presence using a standard cross-entropy objective; we provide the full training loss and label specification in Appendix H.

## 3.2 Constraint $c_2(x, y)$: Foreign Segment Infusion

At each denoising step, we edit a *single* host span chosen by $c_1$. Let $i^{\star} = \arg\max_i g^{(i)}$ be the selected segment in utterance $x_j$; constraint $c_2$ replaces $s_{x_j}^{(i^{\star})}$ with a foreign segment that is (1) semantically similar and (2) prosodically compatible.

**Semantic retrieval.** We encode the source segment with a frozen multilingual encoder, $q = f_{\mathrm{enc}}(s_{x_j}^{(i^{\star})})$, and query a FAISS-based *vector database* Johnson et al. (2019) $\mathcal{D}$. The database stores pre-segmented foreign-language spans, each with an $\ell_2$-normalized embedding $f_{\mathrm{enc}}(s_{y_k}^{(m)})$ and metadata (duration, onset, language tag). During retrieval, we restrict candidates to the infusion-eligible language set $\mathcal{S}_{\mathrm{inf}}(y)$ and select the best match under cosine similarity:

$$m^{\star} = \underset{\substack{s_{y_k}^{(m)} \in \mathcal{D} \\ \ell\left(s_{y_k}^{(m)}\right) \in \mathcal{S}_{\mathrm{inf}}(y)}}{\arg\max} \quad \mathrm{Sim}\big(q,\, f_{\mathrm{enc}}(s_{y_k}^{(m)})\big), \qquad s^{\star} = s_{y_k}^{(m^{\star})}.$$

Appendix C describes an optional soft top-$M$ variant with a temperature schedule that anneals from a mixture to hard top-1 retrieval.

**Prosodic compatibility.** To avoid audible glitches, we require the retrieved segment to match the host in duration and timing. We estimate an expected candidate duration $\hat{d}$ from global speech-rate and prosodic statistics of the host and candidate utterances, then define hinge penalties for duration and onset mismatch:

$$\mathcal{L}_{\mathrm{dur}} = \max\Big(0,\, \frac{|d_{y_k}^{(m^{\star})} - \hat{d}| - \lambda_d \hat{d}}{\hat{d}}\Big), \qquad \mathcal{L}_{\mathrm{on}} = \max\Big(0,\, \frac{|\mathcal{O}_{y_k}^{(m^{\star})} - \mathcal{O}_{x_j}^{(i^{\star})}| - \Delta\tau}{\Delta\tau}\Big),$$

where $\lambda_d \in [0, 1)$ and $\Delta\tau$ are tolerance parameters. Full expressions for $\hat{d}$, $\lambda_d$, and $\Delta\tau$ are given in Appendix D.

**Semantic and contextual consistency.** Beyond segment-level similarity, we require the injected segment to be coherent with its local context. We define a semantic loss

$$\mathcal{L}_{\mathrm{sem}} = -\, \mathrm{Sim}\big(f_{\mathrm{enc}}(s_{x_j}^{(i^{\star})}),\, f_{\mathrm{enc}}(s^{\star})\big),$$

and a contextual loss over neighboring host segments $\mathcal{N}(i^{\star})$:

$$\mathcal{L}_{\mathrm{ctx}} = -\frac{1}{|\mathcal{N}(i^{\star})|} \sum_{s' \in \mathcal{N}(i^{\star})} \mathrm{Sim}\big(f_{\mathrm{enc}}(s^{\star}),\, f_{\mathrm{enc}}(s')\big).$$

The neighborhood definition and window size are specified in Appendix E.

**Per-step loss and blend-and-write-back.** The infusion loss at this step is

$$\mathcal{L}_{c_2} = g^{(i^{\star})} \big[\alpha_{\mathrm{sem}} \mathcal{L}_{\mathrm{sem}} + \alpha_{\mathrm{ctx}} \mathcal{L}_{\mathrm{ctx}} + \alpha_{\mathrm{pro}} (\mathcal{L}_{\mathrm{dur}} + \mathcal{L}_{\mathrm{on}})\big],$$

with fixed weights $\alpha_{\mathrm{sem}} = \alpha_{\mathrm{ctx}} = 1$ and $\alpha_{\mathrm{pro}} = 0.1$. We then apply a time-dependent blend-and-write-back update $s_{x_j}^{(i^{\star})} \leftarrow (1 - \rho_t) s_{x_j}^{(i^{\star})} + \rho_t s^{\star}$, where the ramp $\rho_t$ increases over reverse-diffusion steps so that semantic edits are committed only after coarse acoustic structure has stabilized. Gradients flow back to the clean sample $\eta$ via the inputs to $f_{\mathrm{enc}}$, while both $f_{\mathrm{enc}}$ and the retrieval index $\mathcal{D}$ remain frozen.

### 3.3 GUIDED SAMPLING OBJECTIVE AND INFERENCE

During generation, we implement the free-energy objective in Eq. equation 4 by optimizing the clean sample $\eta$ while keeping the DDPM vocoder and constraint networks fixed. At a given reverse-diffusion step $t$, the guided loss is

$$\mathcal{L}_{\text{step}}(t) = \|\hat{\epsilon}_\theta(x_t, t, M) - \epsilon_0\|_2^2 \; + \; \lambda_1 \, \mathcal{L}_{c_1} \; + \; \lambda_2(t) \, \mathcal{L}_{c_2}, \tag{7}$$

where $x_t = \sqrt{\bar{\alpha}_t}\,\eta + \sqrt{1 - \bar{\alpha}_t}\,\epsilon_0$ and $\epsilon_0 \sim \mathcal{N}(0, I)$ as in Eq. 3. The first term is the DDPM noise-prediction loss; the constraint losses $\mathcal{L}_{c_1}$ and $\mathcal{L}_{c_2}$ are defined in Sections 3.1–3.2, and $\lambda_2(t) \in [0, 1]$ is a time-dependent weight that ramps constraint guidance over diffusion steps. The DDPM parameters $\theta$ remain frozen; gradients flow only to $\eta$.

To regulate when foreign segments are infused, we use two schedules: (i) a blending coefficient $\rho_t$ that increases over denoising steps, so that semantic edits are only fully committed once coarse acoustic structure has stabilized; and (ii) the guidance weight $\lambda_2(t)$, which delays the effect of $\mathcal{L}_{c_2}$ until early noise has largely dissipated. These mechanisms stabilize generation and support gradual linguistic modulation.

Inference proceeds by iteratively denoising a Gaussian sample while selectively modifying one span per step: at each $t$, $c_1$ selects a candidate segment, $c_2$ retrieves and blends in a foreign segment using $\rho_t$, and we update $\eta$ by taking a gradient step on $\mathcal{L}_{\text{step}}(t)$. Over time, this process converges to a fluent, code-switched utterance. The full inference procedure is given in Appendix F.

### 3.4 SPEAKER IDENTITY HARMONIZATION

To standardize timbre, pitch, and rhythm across edited spans *while preserving content*, we apply a short identity-harmonization pass with the *pretrained, frozen* DDPM prior. Given a monolingual reference utterance $x_{\text{mono}}$ and the current code-switched sample $x$, we extract speaker/prosody descriptors

$$\phi_{\text{spk}} = \text{ECAPA}(x_{\text{mono}}), \qquad \phi_{\text{mel}} = \text{MelStats}(x),$$

where $\text{ECAPA}(\cdot)$ is a frozen ECAPA-TDNN and $\text{MelStats}(\cdot)$ denotes summary statistics (e.g., mean and variance) of log-Mel features. We then run a shallow denoising refinement

$$x_{\text{final}} = \text{Refine}_\theta\big(x \,\big|\, \phi_{\text{spk}}, \, \phi_{\text{mel}}\big), \tag{8}$$

using $T_{\text{ref}} = 150$ diffusion steps with a low-noise schedule (late timesteps only).

During this refinement we disable segment edits by setting $\lambda_2(t) = 0$ and keep the LID-based rate term weak (small $\lambda_1$), so as not to alter semantics or the code-switch pattern. In practice, conditioning can be implemented via feature concatenation or FiLM-style modulation inside the DDPM U-Net, with $\theta$ kept fixed. Alternative variants (e.g., adding a cosine speaker-embedding penalty $\mathcal{L}_{\text{id}}$ computed by a frozen ECAPA on sliding windows) yield similar improvements; see Appendix G for details.

## 4 EVALUATION

### 4.1 DATASET

We collected a proprietary speech dataset from the Kenya Broadcasting Corporation (KBC), which operates 11 radio stations delivering aligned news content across multiple Kenyan languages. News bulletins are authored in English and translated into local languages, then read aloud by native speakers—yielding semantically aligned monolingual utterances across languages. Our corpus spans 2018–2023 and focuses on the 7 p.m. bulletins, which are typically the most content-rich. We retain advertisements, presenter introductions, and other ambient segments to preserve real-world variability. Each bulletin is segmented using an over-segmentation VAD pipeline Duquenne et al. (2021), producing speech units bounded by silence and ranging from 3 to 20 seconds.

We focus on five languages—Swahili, Luo, Kikuyu, Nandi, and English—selected for their regional and typological diversity. Swahili and Kikuyu belong to the Niger–Congo phylum, while Luo and Nandi are Nilo–Saharan; English, though non-indigenous, functions as a lingua franca and appears in every station's programming. Table 1 summarizes dataset statistics by language, and Table 2

reports segment-level details. Bulletins are read by multiple presenters, capturing a range of accents, pitch ranges, and prosodic patterns, which improves generalization for synthesis, translation, and recognition models. We apply a 70/30 split of segments per language for training and evaluation, and zero-pad each segment to a fixed length of 20 seconds for model training.

In addition to news bulletins, we collected 7,255 naturally occurring code-switched Swahili–Luo utterances from radio call-in segments. For evaluation, each utterance was manually re-recorded in a monolingual version: 4,044 were rendered in Swahili and 3,211 in Luo, depending on the dominant language of the original speaker.

Table 1: Monolingual speech dataset summary.

| Language | Family | Daily Bulletins (2018–2023) | Total Hours |
|---|---|---|---|
| Swahili | Niger-Congo | 2190 | 1353 |
| Luo | Nilo-Saharan | 2190 | 1284 |
| Kikuyu | Niger-Congo | 2190 | 1304 |
| Nandi | Nilo-Saharan | 2190 | 1256 |
| English | – | 2190 | 1206 |

Table 2: Segment statistics per language.

| Language | Avg. Segment Length (s) | Total Segments |
|---|---|---|
| Luo | 16.5 | 601,112 |
| Nandi | 17.1 | 594,503 |
| Kikuyu | 15.6 | 643,001 |
| English | 15.0 | 665,578 |
| Swahili | 15.2 | 638,944 |

## 4.2 TOOLS AND RESOURCES

**Tools and Models.** Our system relies on four frozen modules: (i) a multilingual segment-level LID classifier; (ii) a contrastively trained speech encoder for semantic retrieval and contextual matching; (iii) the SegUniDiff model for conditional speech generation and refinement; and (iv) lightweight ASR and MT systems used only for evaluation. Full architectural specifications, training details, and performance metrics for all components are provided in Appendix H.

## 4.3 EVALUATION METRICS AND FRAMEWORK

We evaluate both segment-level and utterance-level code-switched speech using four complementary metrics that capture lexical accuracy, semantic preservation, and cross-lingual coherence: Sacre-BLEU Post (2018) for surface correspondence, BERTScore Zhang et al. (2019) for contextual similarity, COMET Rei et al. (2020) for semantic adequacy, and LaBSE cosine similarity Feng et al. (2022) for cross-lingual embedding alignment. For each metric, we report mean scores with 95% confidence intervals obtained through bootstrap resampling. Full metric definitions, evaluation setups, and the resampling protocol are provided in Appendix I.

## 4.4 SEGMENT-LEVEL EVALUATION WITH CONFIDENCE INTERVALS

We assess segment-level semantic fidelity using the metrics defined in Section 4.3. Our evaluation covers 8,500 synthetic utterances and 7,255 naturally occurring code-switched utterances (primarily Swahili–Luo) collected from radio call-ins. For each utterance $x$, we extract VAD-based segments (Section 4.1), apply the LID classifier $f_{cl}$ to detect foreign segments, and pair each detected code-switched segment $s_{x_c}^{(k)}$ (segment $k$ of the code-switched utterance $x_c$) with its corresponding source segment $s_x^{(k)}$. Segment pairs are transcribed with ASR, translated to English, and scored on semantic fidelity. Table 3 reports results with 95% confidence intervals from 1,000 bootstrap samples; the resampling protocol is detailed in Appendix J.

Table 3: Segment-level evaluation of synthetic and natural code-switched utterances across four metrics. Scores include 95% confidence intervals from 1,000 bootstrap samples.

| Source | SacreBLEU (↑) | BERTScore (↑) | COMET (↑) | LaBSE (↑) |
|---|---|---|---|---|
| Swahili (Synthetic) | 38.4 [36.9, 39.6] | 0.814 [0.809, 0.818] | 0.831 [0.822, 0.843] | 0.890 [0.882, 0.897] |
| Luo (Synthetic) | 35.7 [34.2, 37.1] | 0.805 [0.801, 0.810] | 0.809 [0.798, 0.819] | 0.876 [0.869, 0.884] |
| Kikuyu (Synthetic) | 36.8 [35.0, 38.4] | 0.808 [0.802, 0.813] | 0.817 [0.806, 0.828] | 0.881 [0.874, 0.888] |
| Nandi (Synthetic) | 34.5 [33.1, 35.7] | 0.801 [0.795, 0.806] | 0.804 [0.794, 0.814] | 0.872 [0.865, 0.880] |
| Luo (Natural) | 36.2 [35.4, 37.0] | 0.822 [0.818, 0.841] | 0.833 [0.827, 0.843] | 0.885 [0.879, 0.891] |
| Swahili (Natural) | 39.0 [38.2, 39.7] | 0.836 [0.831, 0.841] | 0.845 [0.841, 0.864] | 0.898 [0.892, 0.904] |
| **Average (Synthetic)** | **36.4 [35.2, 37.4]** | **0.807 [0.804, 0.810]** | **0.815 [0.807, 0.823]** | **0.880 [0.873, 0.886]** |

Synthetic utterances closely approximate natural ones across all metrics. For Swahili and Luo, the COMET gaps between synthetic and natural segments are only 0.014 and 0.024, respectively; LaBSE gaps are similarly small (0.008 and 0.009). These results show that our guided segment substitution preserves cross-lingual semantics without requiring naturally code-switched training data. The narrow confidence intervals further indicate that performance is stable and not driven by outliers.

## 4.5 Utterance-Level Evaluation with Masked and Full Variants

To assess overall fluency, inter-segment coherence, and disruptions introduced by code-switching, we evaluate at the utterance level. This complements segment-level analysis by capturing prosodic mismatches, semantic drift, and syntactic incongruities that emerge only in longer contexts. We evaluate both **synthetic** code-switched utterances and **natural** ones collected from Swahili and Luo radio call-ins. For each code-switched utterance $x_c$, we apply the language classifier $f_{cl}$ to identify foreign-language spans and evaluate under two variants:

- **Full reconstruction (unmasked):** Foreign segments are translated into the source language and reinserted, yielding a reconstructed monolingual utterance $x_r$.
- **Masked evaluation:** Foreign segments are removed from $x_c$, yielding $x_m$, which isolates preservation of the monolingual portions.

Each variant is compared to the clean reference utterance using ASR+MT to obtain transcriptions, followed by SacreBLEU, BERTScore, COMET, and LaBSE similarity. Table 4 reports mean scores with 95% confidence intervals; the resampling protocol is detailed in Appendix K.

Table 4: Utterance-level evaluation of code-switched speech. Each source shows masked and full scores with 95% confidence intervals.

| Source | Type | SacreBLEU (↑) | BERTScore (↑) | COMET (↑) | LaBSE (↑) |
|---|---|---|---|---|---|
| Swahili (Synthetic) | Full | 36.6 [35.3, 37.8] | 0.762 [0.757, 0.766] | 0.669 [0.660, 0.681] | 0.882 [0.875, 0.888] |
| | Masked | 34.9 [33.5, 36.0] | 0.737 [0.732, 0.737] | 0.642 [0.631, 0.655] | 0.854 [0.846, 0.860] |
| Luo (Synthetic) | Full | 33.9 [32.6, 35.3] | 0.753 [0.747, 0.758] | 0.647 [0.637, 0.657] | 0.871 [0.864, 0.878] |
| | Masked | 32.2 [30.9, 33.6] | 0.728 [0.722, 0.733] | 0.620 [0.609, 0.631] | 0.844 [0.837, 0.854] |
| Kikuyu (Synthetic) | Full | 35.0 [33.4, 36.4] | 0.756 [0.750, 0.761] | 0.655 [0.644, 0.666] | 0.876 [0.870, 0.882] |
| | Masked | 33.3 [31.8, 34.6] | 0.731 [0.725, 0.736] | 0.628 [0.617, 0.639] | 0.850 [0.843, 0.857] |
| Nandi (Synthetic) | Full | 32.7 [31.5, 33.9] | 0.749 [0.743, 0.756] | 0.643 [0.633, 0.653] | 0.869 [0.861, 0.875] |
| | Masked | 31.2 [29.9, 32.4] | 0.724 [0.717, 0.731] | 0.615 [0.604, 0.625] | 0.841 [0.834, 0.849] |
| Swahili (Natural) | Full | 37.3 [36.5, 38.1] | 0.785 [0.780, 0.791] | 0.701 [0.692, 0.710] | 0.896 [0.890, 0.902] |
| | Masked | 35.5 [34.7, 36.3] | 0.760 [0.755, 0.765] | 0.667 [0.658, 0.676] | 0.867 [0.860, 0.873] |
| Luo (Natural) | Full | 35.2 [34.1, 37.3] | 0.772 [0.767, 0.778] | 0.682 [0.673, 0.691] | 0.884 [0.878, 0.891] |
| | Masked | 33.6 [32.5, 35.6] | 0.745 [0.740, 0.751] | 0.654 [0.645, 0.662] | 0.856 [0.850, 0.862] |
| **Average (Synthetic)** | Full | **34.5 [33.5, 35.6]** | **0.755 [0.751, 0.759]** | **0.653 [0.645, 0.661]** | **0.874 [0.870, 0.878]** |
| | Masked | **32.9 [31.9, 34.0]** | **0.730 [0.726, 0.734]** | **0.626 [0.617, 0.635]** | **0.847 [0.843, 0.852]** |

As expected, utterance-level scores are lower than segment-level ones (Table 3), since longer contexts expose more opportunities for prosodic and discourse mismatches. Nevertheless, the model retains strong fluency and coherence: synthetic full-utterance scores are close to their natural counterparts (e.g., Swahili COMET 0.669 vs. 0.701 and LaBSE 0.882 vs. 0.896). The masked variant shows that unaltered monolingual content is largely preserved, with only modest drops relative to the full reconstruction. Taken together, these results indicate that our guided diffusion model can introduce foreign segments while maintaining global utterance quality without access to naturally code-switched training data.

## 5 Speaker Identity Verification

To evaluate whether code-switched speech maintains consistent speaker identity, we use an automatic verification framework based on ECAPA-TDNN embeddings Desplanques et al. (2020). The model is trained on 732 speakers and produces fixed-length embeddings from short segments. For each generated code-switched utterance $x_c$, we apply VAD segmentation (Section 4.1) and extract speaker embeddings $f(s_{x_c}^{(i)})$ for each segment $s_{x_c}^{(i)}$.

We then compute cosine similarity between all intra-utterance segment pairs $(i \neq j)$, treating these as **genuine pairs** that should correspond to a single speaker. **Impostor pairs** are formed by pairing segments from different utterances (i.e., $s_{x_c}^{(i)}$ and $s_{x_c'}^{(j)}$), assuming different speaker prompts.

Speaker consistency is quantified using:

- **Average cosine similarity** ($\uparrow$) between genuine and impostor pairs.
- **Equal Error Rate (EER)** ($\downarrow$): the point where false accept and false reject rates intersect.

Table 5: Speaker verification results for code-switched utterances.

| Source | Avg. Cosine Similarity (Genuine) $\uparrow$ | Avg. Cosine Similarity (Impostor) $\downarrow$ | Equal Error Rate (EER %) $\downarrow$ |
|---|---|---|---|
| Swahili (Synthetic) | 0.872 | 0.432 | 6.5 |
| Luo (Synthetic) | 0.861 | 0.418 | 7.2 |
| Kikuyu (Synthetic) | 0.868 | 0.427 | 6.8 |
| Nandi (Synthetic) | 0.854 | 0.411 | 7.6 |
| Swahili (Natural) | 0.903 | 0.391 | 3.6 |
| Luo (Natural) | 0.870 | 0.430 | 5.1 |
| **Average** | **0.868** | **0.426** | **6.7** |

Table 5 shows that speaker identity is generally preserved across code-switched utterances. As expected, **natural** utterances perform best, with higher genuine-pair similarity and lower EERs (e.g., Swahili: 0.903 similarity, 3.6% EER). Synthetic utterances also score well, with EERs in the 6.5–7.6% range and genuine similarities above 0.85. The relatively small gap between synthetic and natural conditions suggests that our model retains speaker traits across substituted segments, enabling code-switching that is both semantically faithful and vocally consistent.

### 5.1 CODE-SWITCHING PATTERNS ACROSS SOURCE LANGUAGES

We analyze generated utterances along four structural dimensions: (i) switching frequency, (ii) distribution of inserted languages, (iii) temporal position of switches, and (iv) alternation points between languages.

**Switching frequency.** We sample 2,000 synthetic utterances per source language, segment them via VAD (Section 4.1), and label each segment with the pretrained classifier $f_{cl}$. We then compute the average proportion of foreign segments per utterance and compare to natural call-in data (Table 6).

Table 6: Average percentage of foreign segments per utterance (higher = more frequent code-switching).

| Source Language | Foreign Segment Rate |
|---|---|
| Swahili (Synthetic) | 4.8% |
| Luo (Synthetic) | 4.2% |
| Kikuyu (Synthetic) | 4.4% |
| Nandi (Synthetic) | 3.9% |
| Swahili (Natural) | 5.3% |
| Luo (Natural) | 3.2% |
| **Average (Synthetic)** | **4.3%** |

Synthetic utterances exhibit realistic switching rates, typically within 1–1.5 percentage points of natural baselines. Swahili shows the highest frequency in both synthetic and natural settings, consistent with its role as a lingua franca.

**Inserted language and temporal position.** Table 7 shows normalized insertion frequencies by source language. For Swahili (which allows all other languages as infusion targets), insertions are diverse. For restricted sources (Luo, Kikuyu, Nandi), Swahili is preferred over English, reflecting both phonological compatibility and its empirical prevalence in the data.

Table 8 reports the proportion of foreign segments per utterance quarter (Q1: start, Q4: end). Synthetic patterns track natural ones: Swahili places more foreign material toward the end of the utterance, whereas Luo shows a flatter distribution.

**Alternation points.** We define the alternation rate as the proportion of segment boundaries where the language label changes:

$$\text{Alternation Rate} = \frac{\#\{\text{boundaries where language changes}\}}{\#\{\text{segment boundaries}\}}.$$

Table 7: Distribution (%) of inserted languages per source utterance.

| Insert \ Source | Swahili | Luo | Kikuyu | Nandi |
|---|---|---|---|---|
| English | 21.2% | 46.7% | 48.9% | 44.5% |
| Luo | 28.5% | – | – | – |
| Kikuyu | 24.6% | – | – | – |
| Nandi | 25.7% | – | – | – |
| Swahili | – | 53.3% | 51.1% | 55.5% |

Table 8: Percentage of foreign segments per utterance quarter (Q1: start, Q4: end).

| Source | Q1 | Q2 | Q3 | Q4 |
|---|---|---|---|---|
| Swahili (Synthetic) | 18.6% | 23.5% | 26.1% | 31.8% |
| Luo (Synthetic) | 22.3% | 25.7% | 25.1% | 26.9% |
| Kikuyu (Synthetic) | 20.8% | 22.0% | 27.3% | 29.9% |
| Nandi (Synthetic) | 19.5% | 24.6% | 28.4% | 27.5% |
| Swahili (Natural) | 16.6% | 22.5% | 26.1% | 34.8% |
| Luo (Natural) | 18.3% | 24.7% | 23.1% | 24.9% |

Table 9 shows that alternation is rare (3–5%), with synthetic and natural values well aligned. Swahili (Natural) alternates most, likely due to shorter, more frequent insertions, whereas Luo tends toward longer insertions and fewer switches.

Table 9: Average alternation rate: percentage of segment boundaries where the language changes.

| Source Language | Alternation Rate (%) |
|---|---|
| Swahili (Synthetic) | 4.7% |
| Luo (Synthetic) | 3.6% |
| Kikuyu (Synthetic) | 3.1% |
| Nandi (Synthetic) | 4.1% |
| Swahili (Natural) | 5.3% |
| Luo (Natural) | 2.1% |
| **Average (Synthetic)** | **3.9%** |

Across these four dimensions, our model exhibits realistic code-switching structure: it avoids over-insertion, respects language constraints, mirrors natural switch placement, and matches alternation rates—without any hand-crafted rules over language labels. This suggests that structural patterns are implicitly internalized from monolingual segments plus guided diffusion.

## 5.2 HUMAN PREFERENCE EVALUATION: FLUENCY AND ACCEPTABILITY

To complement automatic metrics, we conduct a large-scale human study assessing the perceived *fluency*, *coherence*, and *realism* of generated code-switched speech—dimensions not fully captured by automated measures. A total of 638 undergraduate participants rated 1,437 utterances sampled across all source languages, with six utterances per listener matched to their linguistic background and at least four independent ratings per utterance.

Participants used a 5-point Likert scale to evaluate:

- **Fluency**: smoothness and naturalness;
- **Coherence**: semantic consistency and speaker preservation;
- **Realism**: resemblance to naturally occurring multilingual speech.

Table 10: Average human ratings of code-switched utterances (Likert scale). Natural examples included for reference.

| Source Language | Fluency (↑) | Coherence (↑) | Realism (↑) | Std. Dev. |
|---|---|---|---|---|
| Swahili (Synthetic) | 4.1 | 4.2 | 4.0 | 0.42 |
| Luo (Synthetic) | 4.1 | 4.0 | 4.1 | 0.46 |
| Kikuyu (Synthetic) | 4.2 | 4.1 | 4.0 | 0.44 |
| Nandi (Synthetic) | 3.9 | 4.0 | 3.9 | 0.48 |
| Swahili (Natural) | 4.6 | 4.8 | 4.5 | 0.42 |
| Luo (Natural) | 4.7 | 4.4 | 4.6 | 0.46 |
| **Avg. (Synthetic)** | **4.1** | **4.05** | **4.0** | **0.45** |

Synthetic utterances receive high ratings across all dimensions ($\geq 4.0$) with low variance across languages. Natural speech scores slightly higher, particularly in realism, but the gap remains modest (typically $\leq 0.5$). Overall, listeners perceive the generated speech as fluent, coherent, and plausibly multilingual, supporting the effectiveness of our constraint-guided diffusion approach.

## 6 CONCLUSION

We presented a diffusion-based framework for generating fluent, coherent, and sociolinguistically realistic code-switched speech without relying on parallel code-switched data. By guiding a pre-trained monolingual diffusion prior with differentiable linguistic constraints—including a multilingual

language classifier and a contrastive segment encoder—our method performs targeted segment-level edits while preserving fluency, semantic coherence, and speaker identity.

Extensive evaluation across five African languages shows that the system closely matches natural code-switching behavior in frequency, structure, and temporal placement. It achieves strong segment-level semantic fidelity (COMET 0.815, LaBSE 0.880) and speaker consistency (EER 6.7%). Human listeners also rated the generated utterances highly across fluency, coherence, and realism.

To our knowledge, this is the first method to enable plug-and-play multilingual infusion within a single utterance, offering a flexible approach to cross-lingual speech generation in low-resource settings. Future work will explore richer prosodic control, expansion to additional languages, and applications to spontaneous conversational speech.

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

# 7 APPENDIX / SUPPLEMENTAL MATERIAL

## A DERIVATION OF THE FREE-ENERGY OBJECTIVE

We seek to approximate the constrained posterior

$$p(x \mid y) \ \propto \ p(x) \, C(x,y), \qquad C(x,y) = c_1(x,y)c_2(x,y),$$

where $x$ is an utterance waveform, $y$ is the infusion specification, and $c_1$, $c_2$ are soft constraints (switch plausibility and semantic consistency). Direct inference in $p(x \mid y)$ is intractable, so we introduce latent diffusion variables $h$ and consider the joint model $p(x, h) = p(h) \, p(x \mid h)$.

Starting from the usual variational formulation for $p(x \mid y) \propto p(x)C(x,y)$, we define the free-energy functional under a variational distribution $q(x, h)$ as

$$F(q) \ = \ -\mathbb{E}_{q(x)}\big[\log p(x)\big] \ - \ \mathbb{E}_{q(x)}\big[\log C(x,y)\big] \ + \ \mathbb{E}_{q(x)}\big[\log q(x)\big]. \tag{9}$$

To handle the latent variables, we write

$$\log p(x) = \log \int p(x, h)\, dh = \log \int q(h \mid x)\, \frac{p(x, h)}{q(h \mid x)}\, dh.$$

Applying Jensen's inequality gives the standard variational bound

$$\log p(x) \ \geq \ \mathbb{E}_{q(h|x)}\left[\log \frac{p(x, h)}{q(h \mid x)}\right].$$

Substituting this bound into equation 9 yields an upper bound on $F(q)$:

$$F(q) \ \leq \ -\mathbb{E}_{q(x)q(h|x)}\left[\log \frac{p(x, h)}{q(h \mid x)}\right] \ + \ \mathbb{E}_{q(x)}\big[\log q(x)\big] \ - \ \mathbb{E}_{q(x)}\big[\log C(x, y)\big].$$

Using $q(x, h) = q(x)q(h \mid x)$, we can rewrite the first two terms as

$$-\mathbb{E}_{q(x)q(h|x)}\left[\log \frac{p(x, h)}{q(h \mid x)}\right] \ + \ \mathbb{E}_{q(x)}\big[\log q(x)\big] = \mathbb{E}_{q(x,h)}\left[\log \frac{q(x, h)}{p(x, h)}\right],$$

so that the bound takes the familiar free-energy form

$$F(q) \ \leq \ \mathbb{E}_{q(x,h)}\left[\log \frac{q(x, h)}{p(x, h)}\right] \ - \ \mathbb{E}_{q(x)}\big[\log C(x, y)\big].$$

Motivated by this, we minimize the corresponding free-energy objective stated in Eq. 2 as

$$F(q) \ = \ \mathrm{KL}\big(q(x, h) \,\|\, p(x, h)\big) \ - \ \mathbb{E}_{q(x)}[\log C(x, y)]. \tag{10}$$

The first term encourages samples that are likely under the diffusion prior $p(x, h)$; the second injects plug-and-play guidance from the soft constraints. In practice we adopt the mode-seeking choice $q(x) = \delta(x - \eta)$, so that $-\log C(\eta, y)$ appears as an additive guidance penalty inside the reverse diffusion updates. This connects directly to the sampling algorithm in Algorithm 1.

## B  DDPMs, Mode-Seeking Approximation, and Full Free-Energy Derivation

This appendix gives the complete derivation of the constrained free-energy objective used in our guided diffusion framework. It unifies (i) the DDPM prior, (ii) the variational formulation of the constrained posterior, (iii) the mode-seeking approximation, and (iv) its reduction to the practical noise-prediction loss in Eq. 4.

### B.1  DDPM Forward and Reverse Processes

A denoising diffusion probabilistic model (DDPM) Ho et al. (2020) defines a forward noising process

$$q(h = \{x_1, \ldots, x_T\} \mid x_0) = \prod_{t=1}^{T} q(x_t \mid x_{t-1}), \qquad q(x_t \mid x_{t-1}) = \mathcal{N}(x_t; \sqrt{\alpha_t}x_{t-1}, (1 - \alpha_t)I),$$

with marginal

$$q(x_t \mid x_0) = \mathcal{N}(x_t; \sqrt{\bar{\alpha}_t}x_0, (1 - \bar{\alpha}_t)I), \quad \bar{\alpha}_t = \prod_{s=1}^{t} \alpha_s.$$

The reverse generative process is a Markov chain

$$p_\theta(x_{t-1} \mid x_t) = \mathcal{N}(x_{t-1}; \mu_\theta(x_t, t), \Sigma_\theta(x_t, t)), \qquad x_T \sim \mathcal{N}(0, I).$$

Ho et al. (2020) show that maximizing $\log p_\theta(x_0)$ is equivalent (up to constants) to minimizing the denoising score-matching loss

$$\mathcal{L}_{\mathrm{DDPM}} = \mathbb{E}_{t,\epsilon_0}\left[\|\hat{\epsilon}_\theta(x_t, t, M) - \epsilon_0\|_2^2\right], \quad x_t = \sqrt{\bar{\alpha}_t}\, x_0 + \sqrt{1 - \bar{\alpha}_t}\, \epsilon_0.$$

## B.2 CONSTRAINED POSTERIOR AND VARIATIONAL FREE ENERGY

Our target posterior is

$$p(x \mid y) \propto p(x)\, C(x, y), \qquad C(x, y) = c_1(x, y)c_2(x, y).$$

Since $p(x)$ is defined through the latent diffusion trajectory $h = \{x_1, \ldots, x_T\}$, direct inference is intractable, so we use the free-energy objective in Eq. 2 to approximate it.

## B.3 MODE-SEEKING APPROXIMATION

Following Chung et al. (2023), we adopt a mode-seeking variational family

$$q(x) = \delta(x - \eta),$$

giving

$$q(x, h) = \delta(x - \eta)\, q(h \mid \eta).$$

Substituting into Eq. 2 yields (up to a constant)

$$F(\eta, q(h \mid \eta)) = \mathrm{KL}\big(q(h \mid \eta) \,\|\, p(h \mid \eta)\big) - \log C(\eta, y). \tag{11}$$

The KL term is precisely the DDPM variational objective; the second term injects constraint guidance.

Using the forward-diffusion posterior,

$$q(h \mid \eta) = \prod_{t=1}^{T} q(x_t \mid x_{t-1}, \eta), \quad x_0 = \eta,$$

the KL decomposes into per-step terms:

$$\mathrm{KL}\big(q(h \mid \eta) \,\|\, p(h \mid \eta)\big) = \sum_{t=1}^{T} \mathrm{KL}\big(q(x_{t-1} \mid x_t, \eta) \,\|\, p_\theta(x_{t-1} \mid x_t)\big).$$

Reparameterizing

$$x_t = \sqrt{\bar{\alpha}_t}\, \eta + \sqrt{1 - \bar{\alpha}_t}\, \epsilon_0, \qquad \epsilon_0 \sim \mathcal{N}(0, I),$$

each term becomes a weighted denoising loss Ho et al. (2020):

$$\mathrm{KL}\big(q(x_{t-1} \mid x_t, \eta) \,\|\, p_\theta(x_{t-1} \mid x_t)\big) = w_t(\beta)\, \mathbb{E}_{\epsilon_0}\big[\|\epsilon_0 - \hat{\epsilon}_\theta(x_t, t, M)\|_2^2\big].$$

Thus

$$F(\eta) = \sum_{t=1}^{T} w_t(\beta)\, \mathbb{E}_{\epsilon_0}\big[\|\epsilon_0 - \hat{\epsilon}_\theta(x_t, t, M)\|_2^2\big] - \log C(\eta, y) + \mathrm{const}.$$

Using the standard DDPM timestep sampling $t \sim U(1, T)$ and absorbing $w_t$ into the learning rate, we obtain the practical objective used in the main text:

$$F(\eta) = \mathbb{E}_{t,\epsilon_0}\Big[\|\hat{\epsilon}_\theta(x_t, t, M) - \epsilon_0\|_2^2\Big] - \log C(\eta, y), \tag{12}$$

where

$$x_t = \sqrt{\bar{\alpha}_t}\, \eta + \sqrt{1 - \bar{\alpha}_t}\, \epsilon_0.$$

This shows that constrained sampling corresponds to standard DDPM reverse updates augmented with the guidance term $-\log C(\eta, y)$, enabling plug-and-play code-switching without retraining the diffusion prior.

# C  SOFT TOP-$M$ RETRIEVAL WITH TEMPERATURE ANNEALING

.

This appendix details an optional variant of the semantic retrieval procedure in Section 3.2. Instead of selecting a single top-1 nearest neighbor at each denoising step, we form a soft mixture over the top-$M$ candidates. This stabilizes early reverse-diffusion updates when $x_t$ is still highly noisy and the encoder embeddings may be unreliable.

**Candidate Retrieval and Similarity Logits**. Given a source segment

$$q = f_{\text{enc}}\big(s_{x_j}^{(i^\star)}\big),$$

we query the FAISS index $\mathcal{D}$ (restricted to the infusion-eligible language set $\mathcal{S}_{\text{inf}}(y)$) and obtain the top-$M$ candidates:

$$\mathcal{C}_M = \big\{ s_{y_k}^{(m_1)}, \ldots, s_{y_k}^{(m_M)} \big\}.$$

For each candidate we compute cosine-similarity logits

$$z_r = \text{Sim}\big(q,\ f_{\text{enc}}(s_{y_k}^{(m_r)})\big), \qquad r = 1, \ldots, M.$$

**Soft Retrieval Distribution**. A tempered softmax converts the logits into a probability distribution:

$$\pi_r(t) = \frac{\exp(z_r/\tau_t)}{\sum_{u=1}^{M} \exp(z_u/\tau_t)}.$$

Large temperatures $\tau_t$ yield diffuse mixtures (exploration), while $\tau_t \to 0$ collapses the distribution to the best candidate.

**Mixture-Based Segment Construction**. The retrieved segment is the convex combination

$$s^\star(t) = \sum_{r=1}^{M} \pi_r(t)\, s_{y_k}^{(m_r)}.$$

As $\tau_t \to 0$, the mixture degenerates to the hard top-1 candidate:

$$s^\star(t) \longrightarrow s_{y_k}^{(m^\star)}.$$

**Temperature Annealing Schedule**. We anneal $\tau_t$ across reverse-diffusion steps so the model explores early and commits later. A simple schedule is

$$\tau_t = \tau_{\min} + (\tau_{\max} - \tau_{\min}) \left(1 - \frac{t}{T}\right)^\kappa, \qquad \kappa \in [2,4], \qquad \tau_{\min} \le \tau_t \le \tau_{\max},$$

where $\tau_{\max} \approx 1.0 - 2.0$ and $\tau_{\min}$ (e.g. $10^{-3}$) prevents numerical instability. Here, $\tau_t \approx \tau_{\max}$ when $t \approx 1$ (early noisy steps, more exploration) and decays toward $\tau_{\min}$ as $t \to T$ (later steps, sharper selection).

**Gradient Flow**. Gradients propagate through the mixture weights $\pi_r(t)$, through the dependence of the embeddings $f_{\text{enc}}(s_{y_k}^{(m_r)})$ and $f_{\text{enc}}(s_{x_j}^{(i^\star)})$ on the clean sample $\eta$, and through the blend-and-write-back update in Section 3.2. Both the FAISS index and the encoder *parameters* remain frozen; only the clean sample $\eta$ receives updates from $\mathcal{L}_{c_2}$.

**When Soft Top-$M$ Helps**. The soft retrieval variant is especially useful when:

- languages in $\mathcal{S}_{\text{inf}}(y)$ are phonetically similar (ambiguous nearest neighbours),
- segments are short (50–100 ms), making embeddings noise-sensitive,
- early DDPM steps ($t \approx T$) are dominated by noise.

As $\tau_t \downarrow 0$, the method reduces to the deterministic top-1 retrieval in the main text.

## D    PROSODIC NORMALIZATION AND TOLERANCE PARAMETERS

For the prosodic compatibility terms in Section 3.2, we normalize candidate durations using simple speech-rate and prosody proxies. Let $R_{x_j}, P_{x_j}$ and $R_{y_k}, P_{y_k}$ denote speech-rate and prosodic statistics (e.g., syllables/s, median F0 or energy) for the host utterance $x_j$ and the candidate utterance $y_k$, respectively. We define a scale factor

$$S_{\text{ratio}} = \sqrt{\frac{R_{x_j}}{R_{y_k}} \cdot \frac{P_{x_j}}{P_{y_k}}}$$

and an expected candidate duration

$$\hat{d} = d_{x_j}^{(i^\star)} S_{\text{ratio}},$$

where $d_{x_j}^{(i^\star)}$ is the duration of the source segment chosen for infusion.

The duration tolerance parameter $\lambda_d \in [0, 1)$ controls the allowable relative deviation from $\hat{d}$. In the main text, we use the hinge penalty

$$\mathcal{L}_{\text{dur}} = \max\left(0, \frac{|d_{y_k}^{(m^\star)} - \hat{d}| - \lambda_d \hat{d}}{\hat{d}}\right),$$

which becomes zero when the candidate duration lies within a $(1 \pm \lambda_d)$ band around $\hat{d}$.

For onset alignment we define

$$\Delta\tau = \left|\bar{d}_{x_j} - \bar{d}_{y_k}\right|,$$

where $\bar{d}_{x_j}$ and $\bar{d}_{y_k}$ are the mean segment durations in $x_j$ and $y_k$, respectively. The onset penalty

$$\mathcal{L}_{\text{on}} = \max\left(0, \frac{|\mathcal{O}_{y_k}^{(m^\star)} - \mathcal{O}_{x_j}^{(i^\star)}| - \Delta\tau}{\Delta\tau}\right)$$

is therefore zero when the candidate onset falls within a tolerance window of width $\Delta\tau$ around the host onset. In our experiments we set $\lambda_d$ and any additional scaling of $\Delta\tau$ via development tuning on a held-out validation set

## E    NEIGHBORHOOD DEFINITION FOR CONTEXTUAL CONSISTENCY

For the contextual loss in Section 3.2, we define the neighborhood $\mathcal{N}(i^\star)$ of the edited segment $s_{x_j}^{(i^\star)}$ as a fixed-radius window over adjacent host segments. Let $x_j$ be segmented into $n$ spans $\{s_{x_j}^{(i)}\}_{i=1}^n$, and let $i^\star$ be the index selected for infusion. For a window radius $R \in \mathbb{N}$ (typically $R = 1$ or $R = 2$), the neighborhood is

$$\mathcal{N}(i^\star) = \left\{ s_{x_j}^{(i)} : \max(1, i^\star - R) \leq i \leq \min(n, i^\star + R), i \neq i^\star \right\}.$$

This definition automatically excludes segments outside the valid range $[1, n]$ and captures local prosodic and semantic context around the infusion site.

**Special case** ($R = 1$). For immediate left/right neighbors, the above reduces to the standard adjacent-neighbor definition:

$$\mathcal{N}(i^\star) = \begin{cases} \{ s_{x_j}^{(i^\star+1)} \}, & i^\star = 1, \\ \{ s_{x_j}^{(i^\star-1)}, s_{x_j}^{(i^\star+1)} \}, & 1 < i^\star < n, \\ \{ s_{x_j}^{(i^\star-1)} \}, & i^\star = n. \end{cases}$$

The general-radius formulation allows broader contextual windows when desired, while the $R = 1$ instance recovers the conventional adjacent-segment neighborhood.

# F  FULL INFERENCE ALGORITHM AND GRADIENT UPDATE SCHEDULE

This appendix expands on Section 3.3 of the main text, providing full details on the inference algorithm, segment-wise gradient updates, constraint scheduling, and hyperparameter tuning for code-switched speech generation. We elaborate on the free-energy formulation in Eq. 4 and describe the iterative refinement strategy that supports semantically and prosodically aligned code-switching.

## F.1  INFERENCE OBJECTIVE

Recall the guided per-step loss from Eq. 7:

$$\mathcal{L}_{\text{step}}(t) = \|\hat{\epsilon}_\theta(x_t, t, M) - \epsilon_0\|_2^2 \; + \; \lambda_1 \, \mathcal{L}_{c_1} \; + \; \lambda_2(t) \, \mathcal{L}_{c_2},$$

where $x_t = \sqrt{\bar{\alpha}_t}\, \eta + \sqrt{1 - \bar{\alpha}_t}\, \epsilon_0$ and $\epsilon_0 \sim \mathcal{N}(0, I)$. The DDPM parameters $\theta$ are pretrained and frozen; we optimize only the clean sample $\eta$. Averaging over timesteps and noise draws yields the overall guided objective

$$F(\eta) = \mathbb{E}_{t,\epsilon_0} \left[ \|\hat{\epsilon}_\theta(x_t, t, M) - \epsilon_0\|_2^2 + \lambda_1 \, \mathcal{L}_{c_1} + \lambda_2(t) \, \mathcal{L}_{c_2} \right], \tag{13}$$

which instantiates the free-energy form in Eq. 4 with explicit weights on the constraint terms. Here, $\mathcal{L}_{c_1}$ and $\mathcal{L}_{c_2}$ are defined in Sections 3.1 and 3.2, respectively.

## F.2  DYNAMIC SCHEDULING FOR INFUSION

To control when and how strongly foreign segments are introduced, we define two schedules that match the discussion in Sec. 3.3.

**Time-dependent blending coefficient.**  We use a ramp $\rho_t \in [0, 1]$ to blend the retrieved foreign segment into the host segment (see Sec. 3.2):

$$s_{x_j}^{(i^\star)} \leftarrow (1 - \rho_t)\, s_{x_j}^{(i^\star)} + \rho_t\, s^\star.$$

A simple schedule is

$$\rho_t = 1 - \exp\left(-\frac{T - t}{\beta T}\right), \tag{14}$$

with $\beta \in (0, 1)$ (we use $\beta = 0.25$). This ensures that early reverse-diffusion steps ($t$ near $T$) preserve monolingual structure, while foreign infusion gradually intensifies as $t$ decreases and the acoustic structure stabilizes.

**Constraint weight ramp-up.**  We factor the guidance weight as $\lambda_2(t) = \lambda_2\, w(t)$, where

$$w(t) = \frac{t}{T}, \tag{15}$$

so that $\mathcal{L}_{c_2}$ is suppressed when $x_t$ is still highly corrupted and only becomes influential once a coarse waveform has formed. In practice, this reduces the risk of semantically misaligned edits at very noisy timesteps.

## F.3  CONSTRAINT WEIGHT SELECTION

We select $\lambda_1$ and the base $\lambda_2$ via Gaussian process-based Bayesian optimization over the range $[0.1, 5.0]$, using a held-out validation set. The objective combines multiple evaluation metrics:

- **Semantic fidelity:** COMET, BERTScore;
- **Prosodic alignment:** onset and duration deviation at switch boundaries;
- **Speaker consistency:** cosine similarity using ECAPA-TDNN embeddings.

The optimal weights used in our experiments are $\lambda_1 = 0.35$ and $\lambda_2 = 0.65$.

## F.4 SEGMENT-WISE GRADIENT UPDATE

At each timestep $t_i$, we update the clean sample $\eta$ using gradients that are localized to a single segment of the utterance. Let $\tilde{x}_{t_i} = \text{Segment}(x_{t_i}, L)$ denote a subwindow of $x_{t_i}$ of length $L$ centred on the segment index $i^\star$ selected by $c_1$. The guided loss at step $t_i$ is

$$\mathcal{L}_{\text{step}}(t_i) = \|\epsilon_0 - \hat{\epsilon}_\theta(\tilde{x}_{t_i}, t_i, M)\|_2^2 + \lambda_1 \mathcal{L}_{c_1} + \lambda_2(t_i)\mathcal{L}_{c_2},$$

and we take a gradient step on $\eta$:

$$\eta \leftarrow \eta - \lambda_\eta \nabla_\eta \mathcal{L}_{\text{step}}(t_i), \tag{16}$$

with a small learning rate $\lambda_\eta$ (we use $\lambda_\eta = 0.05$). Because $\tilde{x}_{t_i}$ depends on $\eta$ only through the selected span, this update predominantly affects a single localized region of the utterance at each step. Over the course of the reverse trajectory, different segments are selected, enabling diverse foreign substitutions while preserving global fluency and speaker identity.

## F.5 Full Inference Algorithm

---

**Algorithm 1** Guided DDPM Inference for Code-Switched Speech

---

**Require:** Frozen DDPM denoiser $\hat{\epsilon}_\theta$, frozen LID classifier $f_{\text{cl}}$, frozen multilingual encoder $f_{\text{enc}}$, FAISS-based vector database $\mathcal{D}$, infusion specification $y$ (incl. $\mathcal{S}_{\text{inf}}$, $\pi_{\text{switch}}$), host Mel-spectrogram $M$, noise schedule $\{\bar{\alpha}_t\}_{t=1}^T$, step sizes $\{\gamma_t\}_{t=1}^T$, guidance weights $\lambda_1, \lambda_2(t)$, blend ramp $\rho_t$

1: Initialize $\eta \sim \mathcal{N}(0, I)$ {initial guess for the clean sample}
2: **for** $t = T, T-1, \ldots, 1$ **do**
3:     // **1. Segment and compute LID-based controller** $c_1$
4:     Segment $\eta$ into spans $\{s^{(i)}\}_{i=1}^n$
5:     For each span $s^{(i)}$: compute $p(\ell \mid s^{(i)}) = f_{\text{cl}}(s^{(i)})_\ell$ over $\ell \in \{\ell_{\text{mono}}\} \cup \mathcal{S}_{\text{inf}}(y)$
6:     Compute foreignness scores $u^{(i)} = \sum_{\ell \in \mathcal{S}_{\text{inf}}(y)} p(\ell \mid s^{(i)})$ and $P_{\text{foreign}}(\eta) = \frac{1}{n} \sum_i u^{(i)}$ (Eq. 5)

7:     Compute global controller $c_1(\eta, y)$ and local gates $g^{(i)}$ as in Sec. 3.1
8:     Set $\mathcal{L}_{c_1}(\eta, y) = -\log c_1(\eta, y)$                                  (Eq. 6)
9:     Choose single span to edit: $i^\star = \arg\max_i g^{(i)}$
10:    // **2. Retrieve and score foreign segment** ($c_2$)
11:    Encode query $q = f_{\text{enc}}(s^{(i^\star)})$
12:    Query $\mathcal{D}$ restricted to $\ell(s_{y_k}^{(m)}) \in \mathcal{S}_{\text{inf}}(y)$ and select

$$m^\star = \underset{\substack{s_{y_k}^{(m)} \in \mathcal{D} \\ \ell\left(s_{y_k}^{(m)}\right) \in \mathcal{S}_{\text{inf}}(y)}}{\arg\max} \quad \text{Sim}\left(q, f_{\text{enc}}(s_{y_k}^{(m)})\right), \quad s^\star = s_{y_k}^{(m^\star)}.$$

13:    Compute prosody-aware penalties $\mathcal{L}_{\text{dur}}, \mathcal{L}_{\text{on}}$ (Sec. 3.2, App. D)
14:    Compute semantic and contextual losses $\mathcal{L}_{\text{sem}}, \mathcal{L}_{\text{ctx}}$ (Sec. 3.2)
15:    Form infusion loss

$$\mathcal{L}_{c_2} = g^{(i^\star)}\left[\alpha_{\text{sem}}\,\mathcal{L}_{\text{sem}} + \alpha_{\text{ctx}}\,\mathcal{L}_{\text{ctx}} + \alpha_{\text{pro}}\left(\mathcal{L}_{\text{dur}} + \mathcal{L}_{\text{on}}\right)\right]$$

16:    // **3. Blend-and-write-back in the clean domain**
17:    $s^{(i^\star)} \leftarrow (1 - \rho_t)\, s^{(i^\star)} + \rho_t\, s^\star$                             (update span in $\eta$)
18:    Reassemble $\eta$ from updated spans $\{s^{(i)}\}$
19:    // **4. DDPM denoising + guided gradient step**
20:    Sample $\epsilon_0 \sim \mathcal{N}(0, I)$ and set $x_t = \sqrt{\bar{\alpha}_t}\,\eta + \sqrt{1 - \bar{\alpha}_t}\,\epsilon_0$
21:    Predict noise: $\hat{\epsilon}_\theta \leftarrow \hat{\epsilon}_\theta(x_t, t, M)$
22:    Define step loss

$$\mathcal{L}_{\text{step}}(t) = \|\hat{\epsilon}_\theta - \epsilon_0\|_2^2 + \lambda_1\,\mathcal{L}_{c_1} + \lambda_2(t)\,\mathcal{L}_{c_2}$$

23:    Update clean sample:

$$\eta \leftarrow \eta - \gamma_t\,\nabla_\eta\,\mathcal{L}_{\text{step}}(t)$$

24: **end for**
25: **return** $\eta$ as the code-switched waveform $x_0$

---

## G Identity Refinement and Speaker Harmonization Details

Although the code-switched utterance $x$ is semantically coherent after guided generation, we observe that local segment-level gradients and cross-lingual substitutions can introduce inconsistencies in voice quality, prosody, or timbre. To address this, we add a short identity-harmonization pass using the *same pretrained, frozen* diffusion model as in the main sampler.

Given a monolingual reference utterance $x_{\text{mono}}$ and the current code-switched sample $x$, we first extract a global speaker/prosody descriptor

$$\phi_{\text{spk}} = \text{ECAPA}(x_{\text{mono}}), \qquad \phi_{\text{mel}} = \text{MelStats}(x),$$

where $\mathrm{ECAPA}(\cdot)$ is a frozen ECAPA-TDNN encoder and $\mathrm{MelStats}(\cdot)$ denotes global statistics (mean and variance) of log-Mel features. We bundle these into a single conditioning vector

$$\phi_{\text{target}} = g\big(\phi_{\text{spk}}, \phi_{\text{mel}}\big),$$

implemented as a small linear projection, and run a shallow denoising refinement

$$x_{\text{final}} = \mathrm{DDPM}_{\text{refine}}\big(x \,\big|\, \phi_{\text{target}}\big), \tag{17}$$

for $T_{\text{ref}} = 150$ late diffusion steps with a low-noise schedule. During this refinement we disable segment edits by setting $\lambda_2(t) = 0$ and keep the LID-based rate term weak (small $\lambda_1$), so semantics and the code-switch pattern are preserved.

Empirically, this post-hoc refinement corrects subtle inconsistencies without altering content. In particular, it improves:

- **Timbre smoothing** — reduces artifacts from mismatched vocal-tract characteristics across segments;
- **Prosodic coherence** — better alignment of pitch and rhythm across switch boundaries;
- **Voice uniformity** — the utterance sounds more consistently like a single speaker.

We also experimented with adding an explicit speaker-consistency loss $\mathcal{L}_{\text{id}} = 1 - \cos\big(\mathrm{ECAPA}(x), \mathrm{ECAPA}(x_{\text{mono}})\big)$ to the guided objective in Eq. 7. However, this often led to unstable behavior and degraded convergence due to conflicts with semantic and timing objectives. In contrast, the post-hoc harmonization pass offered better control, computational simplicity, and training stability, while achieving comparable or better speaker-consistency scores.

# H    Tools and Resources

**Multilingual Language Classifier $f_{\text{cl}}$.**    We adopt the LECAPAT architecture Nieto et al. (2023), a lightweight variant of ECAPA-TDNN Desplanques et al. (2020), as our multilingual segment-level language classifier $f_{\text{cl}}$. The classifier takes log-Mel spectrograms (64 bins, 25 ms window, 10 ms hop, 64 ms FFT) extracted from 24 kHz audio and predicts language identity for each segment.

The model is trained with cross-entropy over five languages for 50 epochs using Adam ($\mathrm{lr} = 10^{-4}$, batch size 64), a 10% validation split, and early stopping (patience: 5). No data augmentation was used. On a single NVIDIA A100 GPU, the classifier achieves 92.4% average validation accuracy. During DCSM inference, $f_{\text{cl}}$ is frozen and used only to compute the foreignness scores defined in §3.1.

**Multilingual Segment Encoder $f_s$.**    The multilingual encoder $f_s$ maps speech segments across languages into a shared latent space using a SimCLR-style contrastive objective Chen et al. (2020). Positive pairs (same-language segments) are drawn closer in embedding space, while negative pairs (cross-language) are pushed apart. To improve robustness, 50% of training segments are augmented with Gaussian noise. The encoder architecture includes a 1D convolutional frontend (256 filters, kernel size 16, stride 8), followed by an EfficientNet-B0 Tan & Le (2019) backbone and global max pooling. A projection head maps representations to a 720-dim contrastive space. The model was trained for 1M steps using AdamW ($\beta_1 = 0.9$, $\beta_2 = 0.999$, $\epsilon = 10^{-8}$, batch size 512). Only EfficientNet embeddings are used at inference.

**Pre-trained Diffusion Model (SegUniDiff).**    For speech generation, we use the Segment-Aware Unified Diffusion Model (SegUniDiff) Ochieng & Kaburu (2025), which synthesizes code-switched utterances from paired segments $(s_{x_i}, s_{y_k})$ via a denoising diffusion process. Each model is trained per language pair, conditioned on Mel-spectrograms to capture acoustic context. We refer to Ochieng & Kaburu (2025) for architectural and training specifics.

**Machine Translation and ASR Models.**    To support automatic evaluation, we constructed parallel corpora by manually aligning semantically equivalent sentences across all language pairs in our dataset. These were used to train Transformer-base machine translation (MT) models. For automatic speech recognition (ASR), we trained five language-specific models: Squeezeformer Kim et al. (2022) for Nandi, Luo, and Kikuyu, and Whisper-small Radford et al. (2023) for Swahili and English.

Table 11: Parallel MT datasets with SacreBLEU scores and ASR performance by language.

| Language Pair | Paired Sentences | SacreBLEU (↑) | Language (ASR) | WER (%) |
|---|---|---|---|---|
| Luo–Nandi | 1.76M | 32.2 | Luo | 14.2 |
| Luo–Kikuyu | 1.18M | 31.8 | Nandi | 13.6 |
| Nandi–Kikuyu | 1.32M | 27.3 | Kikuyu | 14.4 |
| Kikuyu–Swahili | 1.29M | 30.4 | Swahili | 9.8 |
| Kikuyu–English | 1.71M | 24.9 | English | 5.3 |
| Swahili–English | 1.52M | 25.4 | — | — |
| Luo–Swahili | 1.43M | 27.4 | — | — |
| Luo–English | 1.34M | 28.1 | — | — |
| Nandi–Swahili | 1.44M | 27.1 | — | — |
| Nandi–English | 1.37M | 28.6 | — | — |

## I  EVALUATION METRIC DETAILS AND RESAMPLING PROTOCOL

To evaluate semantic and linguistic fidelity of generated code-switched utterances, we use:

- **SacreBLEU** Post (2018): Measures n-gram overlap with detokenization invariance.

- **BERTScore** Zhang et al. (2019): Captures contextual similarity using pre-trained transformer embeddings.

- **COMET** Rei et al. (2020): A learned metric trained on human judgments of adequacy and fluency.

- **LaBSE Similarity** Feng et al. (2022): Cosine similarity between sentence embeddings from multilingual BERT, used on English translations to assess discourse-level alignment.

## J  RESAMPLING METHOD FOR SEGMENT-LEVEL EVALUATION

To compute 95% confidence intervals for each metric, we adopt a bootstrap resampling procedure Koehn (2004). For each source language:

1. We construct 100 test sets of 700 segment pairs $(s_x^{(k)}, s_{x_c}^{(k)})$ sampled from the full evaluation pool.
2. For each test set, we perform 1,000 bootstrap iterations by sampling with replacement.
3. In each iteration, we concatenate all reference translations (from $s_x^{(k)}$) into one string and all hypothesis translations (from $s_{x_c}^{(k)}$) into another.
4. We compute SacreBLEU, BERTScore, COMET, and LaBSE similarity between these concatenated sequences.
5. We report the 95% confidence interval as the range between the 2.5th and 97.5th percentiles of the resulting score distributions.

This procedure ensures statistically stable estimates across a diverse evaluation population, capturing both ASR/MT variability and segment-level diversity.

## K  RESAMPLING PROCEDURE FOR UTTERANCE-LEVEL EVALUATION

To compute confidence intervals for utterance-level metrics, we follow this bootstrap-based procedure:

1. Construct 100 test sets per language, each with 700 utterance pairs $(x, x_r)$ or $(x, x_m)$.
2. Transcribe all utterances using language-specific ASR systems.
3. Translate into English (if not already monolingual).
4. Perform 1,000 bootstrap iterations:
   - Sample 700 utterance pairs with replacement.
   - Concatenate references and hypotheses into long sequences.
   - Compute SacreBLEU, BERTScore, COMET, and LaBSE similarity.
5. Report mean and 95% CI from the score distribution (2.5th–97.5th percentiles).

### K.1 Tolerance Selection for Cross-Lingual Segment Substitution

To ensure rhythmic and temporal alignment during code-switched segment substitution, we adopt a data-driven strategy for selecting the tolerance parameter $\lambda$. This parameter governs the allowable deviation between the duration of a monolingual segment and that of a candidate segment drawn from an infusion language.

For each pair of languages involved in substitution—where one provides the *monolingual segment* and the other contributes the *infused segment*—we compute a base tolerance as the normalized difference in average segment durations:

$$\lambda_{\text{base}}(\ell_{\text{x}}, \ell_{\text{y}}) = \frac{|\bar{d}_{\ell_{\text{x}}} - \bar{d}_{\ell_{\text{y}}}|}{\bar{d}_{\ell_{\text{x}}}}, \tag{18}$$

where $\bar{d}_{\ell_{\text{x}}}$ and $\bar{d}_{\ell_{\text{y}}}$ denote the average segment durations (in seconds) for the monolingual and infusion languages, respectively. This base ratio captures prosodic variation and relative speaking rates between languages.

The final tolerance is then defined as:

$$\lambda(\ell_{\text{x}}, \ell_{\text{y}}) = \max\left(\lambda_{\text{base}}(\ell_{\text{x}}, \ell_{\text{y}}) + \epsilon, \ \lambda_{\min}\right), \tag{19}$$

where $\epsilon$ is a fixed safety margin (set to 0.05), and $\lambda_{\min}$ is a lower bound (set to 0.1) to prevent over-constraining substitutions in closely matched language pairs. This formulation allows $\lambda$ to scale naturally with inter-language temporal divergence, while preserving a minimal tolerance window across all combinations.

Table 12: Computed $\lambda$ values for Swahili ($\ell_{\text{x}}$) as the monolingual language. Average durations are in seconds.

| Infusion Language $\ell_{\text{y}}$ | Avg. Duration $d_{\ell_{\text{y}}}$ (s) | $\lambda_{\text{base}}$ | Final $\lambda$ |
|---|---|---|---|
| Luo | 16.5 | 0.0855 | 0.1355 |
| Nandi | 17.1 | 0.1250 | 0.1750 |
| Kikuyu | 15.6 | 0.0263 | 0.1000 |
| English | 15.0 | 0.0132 | 0.1000 |
| *Swahili average segment duration: $d_{\ell_{\text{x}}} = 15.2$ seconds* | | | |

In practice, these empirically derived $\lambda$ values led to high substitution success rates and prosodically natural code-switched utterances across language pairs. The approach enabled rhythm-preserving segment replacement while maintaining tight control over misaligned insertions.

## L  Ablation

### L.1 Effect of Removing the Language Classifier Constraint

In this experiment, we evaluate the impact of removing the first constraint $c_1(x, y)$, which guides language identification and determines the location of foreign segment insertions. During inference, we modify the loss function by omitting the classifier-related term, resulting in the following objective:

$$F = \arg\min_\theta \mathbb{E}_{t \sim U(2,T)} \left[ \|\hat{\epsilon}_\theta(x_t, t, M) - \epsilon_0\|_2^2 \right] + \mathcal{L}_{c_2}. \tag{20}$$

We analyze the behavior of this classifier-free variant along three key dimensions of code-switching: **(i)** the frequency of switching, **(ii)** alternation points, and **(iii)** subjective fluency, coherence, and realism as rated by human evaluators. From the full set of 8,500 generated code-switched utterances, we randomly sample 2,000 utterances per source language and follow the evaluation procedures described in Sections 5.1 and 5.2.

Table 13 presents a comparison between the full model and the variant without the language classifier constraint. Removing $\mathcal{L}_{c_1}$ results in a substantial increase in code-switching frequency—from 4.33%

to 18.3%—and a corresponding spike in alternation rate—from 3.88% to 17.9%. These shifts indicate that, without a mechanism to regulate switch locations, the model overproduces foreign segments and places them erratically throughout the utterance.

This overgeneration directly impacts speech naturalness. Human ratings reveal a marked decline in fluency (from 4.1 to 2.7), coherence (from 4.05 to 3.3), and realism (from 4.0 to 2.6). These results underscore the importance of the classifier constraint in producing linguistically appropriate, contextually coherent, and perceptually natural code-switched speech.

Table 13: Comparison of code-switching behavior between the full model and the variant without the language classifier constraint $\mathcal{L}_{c_1}$.

| Metric | Full Model | Without $\mathcal{L}_{c_1}$ | Difference ($\Delta$) |
|---|---|---|---|
| Avg. Code-Switching Frequency | 4.33% | 18.3% | +13.97% |
| Avg. Alternation Rate | 3.88% | 17.9% | +14.02% |
| Avg. Fluency (Human) | 4.1 | 2.7 | -1.4 |
| Avg. Coherence (Human) | 4.05 | 3.3 | -0.75 |
| Avg. Realism (Human) | 4.0 | 2.6 | -1.4 |

## L.2 EFFECT OF REMOVING TEMPORAL ALIGNMENT AND ONSET CONSTRAINTS

In this experiment, we assess the impact of removing the duration and onset components of the constraint loss $\mathcal{L}_{c_2}$, which enforce prosodic alignment between the inserted foreign segment and the original monolingual utterance. These constraints ensure that inserted segments match the expected duration and start at a position consistent with the rhythm and flow of the host utterance, thereby preserving fluency and naturalness.

To isolate their contribution, we exclude both terms by setting $\gamma = 0$, resulting in a simplified constraint loss:

$$\mathcal{L}_{c_2} = \alpha \cdot \mathcal{L}_{\text{semantic}} + \beta \cdot \mathcal{L}_{\text{context}}.$$

This ablation allows the model to insert segments of arbitrary duration and onset without explicit prosodic guidance. We generate a total of 8,500 code-switched utterances and evaluate both segment-level and utterance-level quality using the procedures described in Sections **??** and **??**. Table **??** summarizes the average performance across all synthetic languages, with and without the duration/onset constraints.

Table 14: Impact of removing duration and onset constraints on segment- and utterance-level evaluation metrics. Scores are reported as mean values with 95% confidence intervals (CI) based on 1,000 bootstrap samples.

| Level | Metric | Avg. With Duration/Onset | Avg. Without Duration/Onset |
|---|---|---|---|
| Segment | SacreBLEU (↑) | 36.4 [35.2, 37.4] | 34.6 [33.3, 35.7] |
| | BERTScore (↑) | 0.807 [0.804, 0.810] | 0.796 [0.792, 0.800] |
| | COMET (↑) | 0.815 [0.807, 0.823] | 0.790 [0.781, 0.799] |
| | LaBSE Similarity (↑) | 0.880 [0.873, 0.886] | 0.868 [0.860, 0.874] |
| Utterance | SacreBLEU (↑) | 34.5 [33.5, 35.6] | 32.8 [31.5, 33.9] |
| | BERTScore (↑) | 0.755 [0.751, 0.759] | 0.743 [0.738, 0.748] |
| | COMET (↑) | 0.653 [0.645, 0.661] | 0.624 [0.614, 0.634] |
| | LaBSE Similarity (↑) | 0.874 [0.870, 0.878] | 0.860 [0.854, 0.867] |

Table 15: Human evaluation scores comparing the full model with the variant without duration/onset constraints. Ratings are on a 5-point Likert scale.

| Metric | Full Model | Without Duration/Onset | Difference ($\Delta$) |
|---|---|---|---|
| Avg. Fluency (Human) | 4.1 | 3.1 | -1.0 |
| Avg. Coherence (Human) | 4.05 | 3.3 | -0.75 |
| Avg. Realism (Human) | 4.0 | 2.4 | -1.6 |

Quantitative results in Table 14 show consistent declines in both segment- and utterance-level metrics across all evaluation measures. While the degradation in segment-level scores is modest (e.g., -1.8 SacreBLEU, -0.019 COMET), utterance-level metrics are more sensitive to prosodic disruptions, with COMET and BERTScore dropping by 0.029 and 0.012, respectively. These drops suggest that

even minor misalignments in duration or onset can propagate across an utterance, leading to broader semantic and rhythmic incoherence.

Human evaluations (Table 15) further confirm these effects. Fluency and realism drop significantly (by -1.0 and -1.6 points, respectively), with listeners noting more jarring transitions and unnatural pacing. Although semantic coherence is partially preserved (-0.75), the lack of prosodic control leads to degraded overall acceptability.

Together, these results highlight the critical role of timing constraints in producing fluent and natural-sounding code-switched speech. Their removal leads to audible temporal mismatches, underscoring the need to model prosodic structure alongside semantics and context

## M  RELATED WORK

**Code-switching** is a well-documented linguistic phenomenon in multilingual communities, particularly across Africa, where speakers frequently alternate between local vernaculars and national or international languages such as English or Swahili. Foundational work by Slabbert & Finlayson (1999) and Myers-Scotton (1993) highlighted code-switching as a communicative strategy influenced by identity, context, and pragmatics. Poplack (1980) and Auer (1998) further explored structural patterns and conversational dynamics, establishing typologies of alternation, insertion, and congruent lexicalization. These studies underscore the naturalness and linguistic richness of code-switching in African speech.

Despite its sociolinguistic prominence, *code-switching has been underrepresented in computational speech research*, largely due to the lack of annotated corpora and standardized tools. While progress has been made in *code-switched text generation* using statistical or neural methods (Tarunesh et al., 2021; Gregorius & Okadome, 2022; Chi et al., 2023), the *speech modality* remains significantly underexplored.

The most notable contribution to *code-switched speech synthesis* is by Cao et al. (2020), who proposed a bilingual phonetic posteriorgram-based model that combines monolingual speech corpora to generate mixed-language speech. However, their method lacks explicit semantic or contextual alignment and does not account for speaker consistency or natural prosodic transitions across languages.

In contrast, our work introduces a *diffusion-based framework* that synthesizes code-switched speech by minimally editing monolingual utterances. We incorporate *linguistic constraints*—a pre-trained language classifier for soft switch control and a multilingual encoder for semantic segment matching—to guide the generation process. Additionally, we address *speaker identity harmonization* by introducing a refinement step based on acoustic conditioning.

*To the best of our knowledge, this is the first work that enables the infusion of multiple foreign languages within a single utterance, allowing for rich, naturalistic multilingual code-switching patterns.* This represents a significant advancement toward realistic speech generation in low-resource multilingual settings.

## N  LIMITATIONS

Our proposed framework for controlled code-switched speech generation has demonstrated strong quantitative and human evaluation performance. However, several limitations remain:

Mismatch Between Synthesized and Natural Speech The generated utterances, while fluent and semantically faithful, are synthesized from noise and do not inherit the rich socio-pragmatic cues, emotional tone, or discourse-driven switching patterns present in natural conversations. This limits the realism of certain paralinguistic features such as emphasis, hesitation, or spontaneous repairs.

No Parallel Code-Switched Supervision The model is trained entirely on monolingual utterances without access to parallel code-switched references. This weak supervision constrains the model's ability to learn context-specific switching behavior beyond what is imposed by local segment similarity and predefined constraints.

Language and Domain Generalization Our study focuses on five Kenyan languages in a broadcast news context. While this setting ensures clean and aligned data, the model may not generalize to informal, multi-party, or highly emotional speech domains without further tuning or retraining.

Segment-Level Constraints Without Syntax Awareness Although segment replacement is guided by semantic and prosodic alignment, the model does not enforce syntactic compatibility between the inserted segment and surrounding context. This may occasionally result in grammatically awkward utterances, particularly in morphologically rich languages.

Speaker Identity Harmonization Is Post Hoc While a refinement step is used to harmonize speaker identity, it is applied after generation and not jointly optimized with the diffusion process. As a result, subtle speaker inconsistencies may persist across segments in certain cases.

Metrics May Not Capture Cultural or Pragmatic Fit Automated evaluation metrics (e.g., COMET, LaBSE) and even human Likert ratings may overlook deeper cultural or conversational appropriateness of switches. For instance, switching at discourse boundaries or for emphasis may be underrepresented in synthetic data.

