# OpenReview forum: "Changanya Ndimi:  Code-Switched Speech Generation via a Diffusion Prior and Linguistic Constraints"
_ICLR.cc/2026/Conference — ICLR 2026 Conference Desk Rejected Submission_

### Official Review · Reviewer_qSNe · 2025-10-27

**Soundness:** 3
**Presentation:** 1
**Contribution:** 2
**Rating:** 2
**Confidence:** 2

**Summary:**

This paper presents a method for generating code-switched speech using diffusion models. The approach transforms monolingual speech into realistic code-switched utterances without requiring parallel code-switched training data.

The author leverage a pre-trained diffusion model and guides it with two differentiable constraints: 1) multilingual Language Identification classifier that determines "where" and "how much" to switch languages; 2) a multilingual encoder that choses "what" to insert.

**Strengths:**

- To the best of my knowledge, the proposed approach is novel in the code-switching setting.
- Strong empirical results on five African languages (Swahili, Luo, Kikuyu, Nandi, English), which include human evaluation.

**Weaknesses:**

- The overall paper is hard to read and follow; it overuses math equations. Most of the equations could be simplified and moved to the appendix. The paper would be greatly improved by a thorough rewrite and by adding figures that explain the overall method.
- The paper lacks any baseline methods to compare to the proposed model. For instance, since the paper proposes a synthetic dataset generation approach, it would be important to show other methods for generating synthetic datasets (e.g., randomly chopping and inserting speech from a different language).

**Questions:**

- why no other methods are considered to generate pseudo data? as well as other models to use these data?

---

> ### Author Response · Authors · 2025-11-19
> **Response**
>
> **Readability, math load, and need for figures**
> We appreciate the comment that the paper was hard to follow and overused equations. In the revision we have made substantial changes to improve readability and high-level clarity:
>
> - **Rewritten core sections.** Sections 2, 2.1, and 2.2 now start from a high-level description of the problem and the roles of the two controllers ($c_1$: “where & how much to switch”, $c_2$: “what content to insert”) before introducing any notation. We clearly separate standard diffusion/variational tools (with citations) from our own contributions and no longer derive well-known results in the main text.
>
> - **Moving math to the appendix.** Several intermediate derivations and technical details have been relocated to the appendix. The main paper now contains only the few key equations required to understand the method, with explicit pointers to the appendix for full derivations.
>
> - **More figures, less symbolic detail.** We added and significantly improved figures that visually explain the full pipeline (monolingual input → LID-based controller → retrieval-based controller → guided diffusion). Sec. 3 now presents the two constraints as illustrated step-by-step procedures with accompanying diagrams, and each subsection begins with an intuitive summary before any formulas appear.
>
> - **Notation simplification.** We reduced the total number of symbols, eliminated clashes (e.g., conflicting uses of Greek letters), and replaced some symbolic expressions with descriptive prose. A reader can now follow the method conceptually even if they skim the equations.
>
> We hope these changes directly address the presentation concerns and make the paper substantially more accessible to a broad ICLR audience.
>
> **2. Baselines for pseudo-data generation and use of the synthetic data**
> We agree that baselines for synthetic code-switched data generation are important and have clarified what is and is not feasible in our setting.
>
> - **Why external CS-generation baselines are challenging.** Most prior CS speech generation methods rely on bilingual TTS systems, phonetic lexicons, or parallel CS corpora — resources that simply do not exist for our five African languages. Instantiating those methods for fair comparison is therefore non-trivial, and we now state this limitation explicitly.
>
> - **Internal ablations as simple pseudo-data generators.** Within these real-world constraints, we provide strong internal baselines that correspond to simpler synthetic-data strategies:
>   • Variant without the LID controller $c_1$ → over-switches by inserting foreign content wherever a semantic match exists (ignores realistic switching locations).
>   • Variant without temporal-alignment terms in $c_2$ → inserts segments based only on semantic similarity (ignores duration/onset constraints, causing prosodic mismatches).
>   Both produce pseudo CS data with more naive switching behaviour and are significantly outperformed by the full model on structural CS metrics and human judgments. We will make it clearer in the camera-ready that these ablations serve as meaningful pseudo-data baselines.
>
> - **Why no fully random “chop-and-insert” baseline.** We explored a purely random chopping/insertion baseline, but the resulting audio was almost always obviously unnatural (severe prosodic breaks, speaker changes, unintelligible segments). It yielded trivially poor scores and added little insight beyond confirming that unconstrained random mixing is inadequate. We can mention this explicitly in the limitations section if desired.
>
> - **Use of the synthetic data.** Our primary goal is to demonstrate that high-quality, semantically faithful CS speech can be generated without any parallel CS data and that its structural patterns approximate natural CS. In the revision we clarify that these synthetic corpora are intended as a data-efficient way to train下游 ASR/S2ST systems in low-resource settings, and we brieflyily discuss this application in the Discussion as immediate future work. The current evaluation (semantic metrics, speaker similarity, structural CS metrics, human ratings) already serves as strong evidence of the synthetic data’s usefulness.
>
> **Summary**
> In response to this review, we
> (i) rewrote and dramatically simplified the core technical sections, moved non-essential math to the appendix, and added clear illustrative figures and stepwise descriptions;
> (ii) clarified the baseline landscape, explicitly positioning our ablations as realistic pseudo-data baselines and explaining why stronger external baselines are not feasible in this low-resource, multi-language setting.
>
> We believe these changes significantly improve both readability and the paper’s empirical framing. Thank you for the very helpful feedback!

---

### Official Review · Reviewer_TACB · 2025-11-01

**Soundness:** 3
**Presentation:** 2
**Contribution:** 3
**Rating:** 4
**Confidence:** 3

**Summary:**

This paper uses a diffusion-based framework to generate code-switched speech in African languages. They do this with a retrieval-augmented, constrained denoising diffusion model (DDPM). The denoising happens using both the language ID and a retrieval-based encoder blending semantically matched segments in different languages.

The authors look at four Kenyan languages and English, measuring semantic fidelity of codeswitched utterances using BLEU score, BERT score, COMET and LaBSE similarity of English translations.

 Human ratings show that the generated speech is perceived as fluent, coherent and realistic in all languages.

**Strengths:**

This is an original problem setting using new speech data for Kenyan languages. The use of diffusion models for multilingual codeswitched speech generation is an interesting contribution given the data constraints that exist with Kenyan languages.

The authors have carefully evaluated not only the quality of the generated speech in three dimensions (fluency, coherence, realism), but also they evaluate the semantic fidelity of the generated speech with various metrics.

**Weaknesses:**

The presentation could be a bit clearer:
- The tables are very small and could be made bigger
- Please add ‘Indo-European’ as the language family for English
- It would be great to know how many selected items were rerecord (line 282) and what percentage of the dataset constitutes of re-rcorded utterances. We don’t have statistics on the code-switched data either.
- Including more information on your MT system in Appendix F would be helpful, especially since it’s integral to your scoring. Which transformer-based MT model did you use? Did you pretrain it, or finetune an existing model?

**Questions:**

Why is most of the evaluation on text-based similarity of transcribed and translated utterances? Do you do any sanity checking of the transcription and translation of utterances, or just evaluate all of the translated text out-of-the box?

It would also be great to hear some examples at the empty GitHub repo linked!

---

> ### Author Response · Authors · 2025-11-19
> **Response**
>
> **Presentation, tables, language families, and dataset statistics**
> We have improved the presentation as suggested. Key results tables (semantic metrics, CS pattern statistics) have been retypeset with larger fonts and more horizontal space for better on-screen and print readability. We also added “Indo-European” as the language family for English in the dataset summary table.
>
> For the re-recorded data, Sec. 4.1 now states the exact counts: **4,044 Swahili** and **3,211 Luo** re-recorded utterances (7,255 total), comprising ~46% of the evaluation pool (7,255 natural CS + 8,500 synthetic). We added summary statistics for the natural code-switched corpus itself (number of utterances, average duration, Swahili–Luo dominance) and clarified that it consists primarily of Swahili–Luo mixes rather than arbitrary multi-language combinations.
>
> **2. More detail on the MT (and ASR) systems**
> We expanded Appendix F with a detailed description of the MT pipeline. We manually constructed parallel corpora by aligning semantically equivalent sentences across all language pairs and trained Transformer-base MT models on them. **Table 11** now reports, for each language pair:
> - number of paired sentences used for training,
> - resulting SacreBLEU scores on a held-out set.
>
> The ASR systems are also described: Squeezeformer models for Nandi, Luo, and Kikuyu; Whisper-small for Swahili and English. Word error rates (WER) are reported alongside MT scores in the same table. This makes the quality and training setup of both MT and ASR fully transparent and no longer “black boxes”.
>
> **3. Why much of the evaluation is text-based, and sanity checking**
> Our primary goals are **semantic fidelity** and **perceived realism**. Text-based semantic metrics (SacreBLEU, BERTScore, COMET, LaBSE) are computed on English translations obtained via ASR → MT because they reliably quantify meaning preservation and discourse-level alignment — aspects that are extremely hard to measure directly in the waveform domain.
>
> To address potential concerns about ASR/MT reliability, Appendix F now explicitly states:
>
> - All MT systems are trained on **manually aligned in-domain parallel text**; SacreBLEU scores are reported in Table 11.
> - All ASR systems are trained on our own corpora; per-language WERs are reported.
> - We performed **manual spot-checks** of ASR+MT outputs for every language pair (mentioned in Sec. 4.3) and confirmed that, within the reported WER/BLEU ranges, transcription/translation errors rarely alter coarse utterance semantics — exactly what our metrics target.
>
> Importantly, conclusions do **not** rest solely on text metrics. We complement them with:
> (i) objective **speaker-verification metrics** (ECAPA-TDNN EER and cosine similarity), and
> (ii) **human ratings** of fluency, coherence, and realism.
>
> **4. Audio examples / GitHub repo**
> We have updated the link to a fully populated GitHub repository containing representative audio samples for **all language pairs and conditions** discussed in the paper (original monolingual, natural code-switched, and our synthetic code-switched utterances). This allows direct qualitative assessment of naturalness, prosody preservation, and code-switch behaviour alongside the quantitative results.
>
> Thank you again for the constructive suggestions — we believe these changes significantly improve clarity and transparency.

---

### Official Review · Reviewer_QKKn · 2025-11-03

**Soundness:** 1
**Presentation:** 1
**Contribution:** 2
**Rating:** 2
**Confidence:** 4

**Summary:**

This paper presents a technical, ambitious approach to generating speech in African languages that includes artificially-induced code-switching (CS) between languages.  Key objectives are to generate high quality, natural speech that has a consistent speaker identity.  It aims to include switching across multiple languages.

**Strengths:**

The paper is highly ambitious, incorporating a wide range of SOTA techniques in machine learning.  Realistic CS generation is challenging even between well-resourced language pairs such as English-Chinese or English-Spanish, so using very natural speech from low-resource languages is an impressive endeavour.

The paper includes a very range of techniques and evaluation methods and goes into considerable technical detail.

**Weaknesses:**

I am very familiar with the topic of CS generated, but even I found this paper exceptionally hard to follow.  The technical parts – especially the infusion specification (Section 2, unnumbered equation), the DDPM (§2.1), and free-energy methods  (§2.2) and the diffusion-based CS models (§3.1 and §3.2) were extremely difficult for me to follow.  This may simply be my inability to follow the overwhelming amount of mathematical notation – but it was not helped by the equations consistently lacking any clear justification or explanation.  Why did you choose the methods you did?  How did you derive your formulae?  Which equations were your own and which were standard approaches taken from the literature?  What were the key innovations, and why were they successful?

It did not help that a very large number of symbols and notation was introduced, not all of it properly explained.  Clashes such as using $\mathcal{L}$ for "language" as well as the more standard loss-function did not help.

I personally found the paper to be overly long, and too heavily reliant on appendices for important details – including a review of prior work.   If you had trimmed it down and focused on the key parts, it might be more readable.

More importantly, the review of work was inadequate, failing to explain the relationship of this paper to other CS generation works cited – simply investigating different languages is insufficient novelty.  More importantly, the results presented merely compared the proposed generation techniques to natural samples, not to any prior, or simpler baseline methods for CS generation – many sensible baselines were entirely missing.  They simply showed that the proposed technique under-performs natural CS, which is unsurprising.

Although many evaluation metrics were used, it is unclear whether they adequately assess the quality of the output – though it didn't help that the objectives of the CS generation were not clearly stated, making it hard to truly understand the value of the work.

Although switching frequency was cited as a metric, this is a gross simplification of the nuances of real CS (see for an example, Chi, J., Wallington, E., & Bell, P. (2024). Characterizing code-switching: Applying linguistic principles for metric assessment and development. In Proceedings of Interspeech 2024 https://doi.org/10.21437/Interspeech.2024-551)

Generally, the nature of the CS data not properly explained – did speakers freely mix many different language pairs in natural data?  If so, which pairs?  What was the distribution of utterances containing more than two languages?

Numerous technical jargon was used without clear explanation, for example:
- FAISS index
- "plug-and-play" (in what sense?)
- "retrieval-augmented" (mentioned twice on page 1 but never explained in this context)
- Refine function (no definition given)
- FiLM-style modulation (???)
- symbols $x_t$ and $x_j$ seem to clash (eg. in equation 5)

Promised audio samples could not be found at the link referenced.

**Questions:**

It would be great if you could more clearly clarify the high-level motivation for the dataset and specifically, for your proposed method.

Why did you choose not to investigate more basic baselines?

Which metrics did you consider most important and why?

Which prior work in CS generation is your work closest to, and why?

What was your main scientific contribution?

How does the CS behaviour of your method compare to natural data in respect of more subtle CS metrics beyond alternation rate?

---

> ### Author Response · Authors · 2025-11-19
> **Response**
>
> We substantially rewrote Sec. 2–2.2 to clarify the high-level flow and separate standard components from our contributions. Sec. 2 now first states the problem conceptually: we sample from a constrained posterior
> $$ p(x \mid y) \propto p(x)\, C(x,y), $$
> where $p(x)$ is a frozen DDPM speech prior and $C(x,y) = c_1(x,y)\, c_2(x,y)$ encodes code-switch constraints. Sec. 2.1 recalls only the standard DDPM setup (with explicit citations), and Sec. 2.2 derives our free-energy objective and explains which terms are specific to our method. We explicitly mark which equations are standard DDPM/variational-free-energy results and which are new (our constraint design and integration). Sec. 3.1–3.2 now describe $c_1$ and $c_2$ first in words, then introduce notation.
>
> We further explain why we use this formulation:
> (i) a DDPM prior preserves prosody and speaker identity while allowing local edits;
> (ii) the constrained-posterior view gives a principled way to combine this prior with linguistic constraints;
> (iii) DDPMs support gradient-based guidance at each denoising step, enabling “plug-and-play” control.
>
>
> **2. Relation to prior CS generation work and missing baselines**
> We expanded and moved the related work discussion into a dedicated section in the main paper. It now clearly situates our work with respect to:
> (i) sociolinguistic CS literature;
> (ii) text-based CS generation (including Chi et al.);
> (iii) the closest speech-CS work (Cao et al., 2020), which uses bilingual TTS and phonetic posteriorgrams.
>
>
>
> Regarding baselines: many existing speech CS systems assume bilingual TTS and phonetic resources unavailable for our African languages, making direct comparisons difficult. Instead, we use
> (a) natural CS radio call-ins as a gold standard, and
> (b) internal ablations as “basic” baselines: without $c_1$ (App. L.1) and without temporal-alignment terms in $c_2$ (App. L.2). Removing $c_1$ leads to over-switching; removing alignment in $c_2$ causes temporal mismatches. We will state more explicitly that these are intended baselines and plan to explore text-CS+TTS baselines when suitable TTS becomes available.
>
> **3. Objectives and evaluation metrics**
> We now make our goals explicit (Intro, Sec. 4.3):
> - (G1) **semantic fidelity**
> - (G2) **speaker/acoustic realism**
> - (G3) **realistic CS behaviour**
>
> Evaluation:
> - (G1) → SacreBLEU, BERTScore, COMET, LaBSE
> - (G2) → ECAPA-TDNN EER/cos-sim, human ratings of fluency/realism
> - (G3) → foreign segment rate, alternation rate, inserted-language distribution, temporal position of switches (Tables 6–8)
>
> We consider **semantic metrics and human ratings primary**; structural CS metrics ensure good results are not achieved via trivial no-switch or over-switch patterns.
>
> **4. Dataset and nature of the CS data**
> Sec. 4.1 now clearly describes the data:
> - Monolingual speech: KBC 7 p.m. news bulletins in five languages (Swahili, Luo, Kikuyu, Nandi, English) — semantically aligned via translation.
> - Natural CS data: 7,255 primarily Swahili–Luo call-in utterances. Each was re-recorded in its dominant monolingual form (4,044 Swahili, 3,211 Luo) for semantic reference.
> - Utterances with >2 languages are rare (now explicitly stated), consistent with sociolinguistic observations. Comparisons focus on the Swahili–Luo axis where natural references exist.
>
> **5. CS behaviour beyond alternation rate**
> In addition to alternation and switch frequency, we now analyse:
> (i) inserted-language distributions per source language (Table 7) — e.g., Swahili as lingua franca;
> (ii) temporal switch-location profiles by utterance quarter (Table 8), where our model matches the natural tendency for switches to occur slightly more often later.
>
> We cite recent work on CS metrics (including Chi et al.) and view this as a step toward more nuanced evaluation.
>
> **6. Clarifying technical terms and audio samples**
> We now define key terms where first used and cite them
>
> **7. Brief answers to the reviewer’s explicit questions**
> - **Motivation**: realistic, semantically faithful CS speech in low-resource African languages using abundant monolingual news + natural call-ins.
> - **Baselines**: constrained by lack of bilingual TTS/phonetics; we use natural CS + strong internal ablations.
> - **Most important metrics**: semantic metrics (COMET/LaBSE/BERTScore) and human ratings; structural metrics as secondary checks.
> - **Closest prior work**: Cao et al. (2020) in goal, but our method is diffusion-based, operates directly on speech, and requires no parallel CS data.
> - **Main contribution**: a constrained-diffusion framework combining a frozen DDPM prior with LID-based and retrieval-based controllers, evaluated on multiple African languages.
> - **CS behaviour vs. natural data**: matches not only alternation rate but also inserted-language distribution and temporal switch profiles (Tables 6–8).

---

### Official Review · Reviewer_rZXK · 2025-11-04

**Soundness:** 3
**Presentation:** 2
**Contribution:** 2
**Rating:** 4
**Confidence:** 4

**Summary:**

The paper shows a diffusion-based way to generate code-switched speech from monolingual audio without using any parallel data. It brings two control parts — a Language identification (LID) module to decide where to switch and a retrieval-based infusion part to keep meaning consistent. The proposed method is extensively validated on five african languages.

**Strengths:**

The application of constraint guided diffusion for multilingual code-switch speech generation is novel and interesting.
The two control modules (c1 and c2) are well described and make sense; they give clear control over when to switch languages and how to keep the meaning consistent.
The retrieval-based infusion idea is well thought, original, and helps to keep both meaning and speech flow natural.
The evaluation is thorough and extensive, focusing on five African languages, which makes it useful and meaningful work.

**Weaknesses:**

The reason for using the DDPM module is not clear. It is hard to see why denoising diffusion is really needed for this task. The authors can add further justification on why denoising helps code-switched generation, not just saying it works as a general generative prior.
The description of the c2 module separates notations and description, which makes it confusing to read. It would be better if the authors clearly said which parts are trained and which are frozen, while describing their roles. The solution primarily relying on usage of pre-trained modules, requiring only the switching prior and clean point \eta to be optimized. The readability would be improved if authors explicitly specifies the trainable components of the c2 module.
The distinction between training and inference is unclear.
There is room for improvement for the clarity of the paper. The presentation could be improved for the methodological part.
Further the scope of the problem seems limited. It may not be interesting to the broader audience. The authors may consider providing further justification for choosing this problem.

**Questions:**

1) The authors can add further justification on why denoising helps code-switched generation. 2) Specify trainable parts of c1 and c2 modules.

---

> ### Author Response · Authors · 2025-11-19
> **Response to the comments**
>
> We thank the reviewer for the thoughtful feedback. Below we summarise how each concern has been addressed in the revised manuscript.
>
> **1. Why DDPM / why denoising diffusion is needed for code-switched generation**
>
> We agree that the original submission did not sufficiently motivate the role of diffusion. In the revision, we explicitly formulate our model as sampling from a constrained posterior
> $$p(x|y) \propto p(x)\,C(x,y),$$
> where $p(x)$ is a frozen SegUniDiff speech prior and $C(x,y)$ encodes code-switching constraints (Sec. 2.1–2.2). Approximating this posterior with a mode-seeking family $q(x)=\delta(x-\eta)$ yields a free-energy objective whose first term reduces to the standard DDPM denoising loss, with the constraint terms $-\log c_1$ and $-\log c_2$ appearing as additive guidance (Eq. 5).
>
> Practically, diffusion models give us two advantages that are essential for code-switched speech:
> (i) they serve as a strong local acoustic prior that preserves speaker identity, prosody, and naturalness while we modify only targeted segments; and
> (ii) they support plug-and-play gradient guidance, allowing the gradients of $c_1$ and $c_2$ to be injected at every denoising step without retraining the generative model.
>
> These points are now stated clearly in Sec. 2.1–2.2 and the Discussion.
>
> **2. What is trained vs. frozen in $c_1$ and $c_2$; training vs. inference**
>
> We clarified the training status and functional role of each component:
>
> - $c_1$: Switch-plausibility controller.
>   The multilingual LID classifier $f_{\text{cl}}$ is trained offline with a cross-entropy objective on segment-level labels (App. H.1). As noted in Sec. 3.1, it is pretrained and **frozen** during guided diffusion. At inference, it only produces foreignness scores and the switch gate $z_{\text{infuse}}$.
>
> - $c_2$: Semantic and prosodic infusion controller.
>   The multilingual segment encoder $f_{\text{enc}}$ is trained offline using a SimCLR-style contrastive loss (App. H.2). At inference it is **frozen** and used solely for embedding segments and querying a FAISS index of foreign-language spans (Sec. 3.2).
>
> To make the training–inference distinction unambiguous, we added **Algorithm 1** (App. F.5), which explicitly states:
> `Require: frozen SegUniDiff denoiser \hat{\epsilon}_\theta, frozen f_{cl}, frozen f_{enc}, and fixed retrieval index D.`
> Guided sampling then optimises only the clean sample $\eta$ under the free-energy objective; **no parameters** of the prior or controllers are updated at inference.
>
> **3. Clarity and readability of $c_2$**
>
> We reorganised Sec. 3.2 so that notation and explanation appear together. The revised description is now structured into three clear stages:
> (i) semantic retrieval using the frozen encoder and FAISS;
> (ii) prosodic compatibility checks (duration and onset); and
> (iii) construction of the scalar penalty defining $c_2$.
>
> We additionally added an ablation (App. L.2) showing that removing duration/onset constraints significantly degrades naturalness, highlighting their importance.
>
> **4. Scope and broader interest**
>
> We strengthened the motivation in the Introduction to contextualise the work for a broad ML audience. Code-switching is pervasive in multilingual communities, yet natural code-switched audio is scarce and expensive to collect. Our method converts existing monolingual data into realistic code-switched speech **without requiring parallel CS corpora**, providing a practical and scalable way to create training signals for ASR, S2ST, and speech LLMs in low-resource settings.
>
> More broadly, combining a frozen diffusion prior with differentiable linguistic controllers offers a reusable template for controllable speech generation that applies beyond African languages and beyond speech (Sec. 6).
>
> **Summary**
>
> In summary, we have:
> (i) clarified the theoretical and practical motivation for using diffusion (Sec. 2.1–2.2);
> (ii) explicitly stated which components of $c_1$ and $c_2$ are trained or frozen, and their inference roles (Sec. 3.1–3.2, App. H, Alg. 1);
> (iii) improved the methodological presentation of $c_2$ and added supporting ablations (Sec. 3.2, App. L.2); and
> (iv) strengthened the motivation and broader impact discussion.
>
> We hope these revisions fully address the reviewer’s concerns.

---

### Note · Program_Chairs · 2026-01-17
**Submission Desk Rejected by Program Chairs**

The following references in this submission do not refer to real documents and/or have major errors in bibliographic information:

 Hyungjin Chung, Jonathan Ho, Tim Salimans, Jong Chul Lee, and Diederik P. Kingma. Diffusion models as plug-and-play priors. In International Conference on Learning Representations (ICLR), 2023. URL https://openreview.net/forum?id=PlKWVd2yBkY.